# Recent Progress and Perspective: Na Ion Batteries Used at Low Temperatures

**DOI:** 10.3390/nano12193529

**Published:** 2022-10-09

**Authors:** Peiyuan Li, Naiqi Hu, Jiayao Wang, Shuchan Wang, Wenwen Deng

**Affiliations:** 1Research Center of Green Catalysis, College of Chemistry, Zhengzhou University, 100 Science Road, Zhengzhou 450001, China; 2Institute of Materials Science & Devices, School of Material Science and Engineering, Suzhou University of Science and Technology, Suzhou 215000, China

**Keywords:** sodium-ion battery, low temperature, all climate, conductivity

## Abstract

With the rapid development of electric power, lithium materials, as a rare metal material, will be used up in 50 years. Sodium, in the same main group as lithium in the periodic table, is abundant in earth’s surface. However, in the study of sodium-ion batteries, there are still problems with their low-temperature performance. Its influencing factors mainly include three parts: cathode material, anode material, and electrolyte. In the cathode, there are Prussian blue and Prussian blue analogues, layered oxides, and polyanionic-type cathodes in four parts, as this paper discusses. However, in the anode, there is hard carbon, amorphous selenium, metal selenides, and the NaTi_2_(PO_4_)_3_ anode. Then, we divide the electrolyte into four parts: organic electrolytes; ionic liquid electrolytes; aqueous electrolytes; and solid-state electrolytes. Here, we aim to find electrode materials with a high specific capacity of charge and discharge at lower temperatures. Meanwhile, high-electrical-potential cathode materials and low-potential anode materials are also found. Furthermore, their stability in air and performance degradation in full cells and half-cells are analyzed. As for the electrolyte, despite the aspects mentioned above, its electrical conductivity in low temperatures is also reported.

## 1. Introduction

Nowadays, with the rapid development of mobile electronic devices and increasing energy consumption, people’s demand for lithium-ion batteries (LIBs) is expanding [1]. However, judging from the current scarcity of lithium resources, both in terms of cost and distribution, LIBs are not the best choice in future market. Na has chemical properties which are similar to lithium [2], so that sodium-ion batteries (SIBs) have high expectations as an alternative to LIBs [3]. Compared with LIBs, SIBs have several advantages. Sodium resources are abundant, and the price is relatively low. They also have the advantages of the low-concentration electrolyte which can be taken to reduce costs [4]. Compared with the same concentration of the lithium-ion electrolyte, the transfer efficiency of sodium ions in the ion electrolyte is 20% higher than that of lithium ions [5]. On the other hand, the collector of lithium-ion batteries is mostly copper foil, as lithium materials are more likely to interact with aluminum metal [6]. In this case, the use of aluminum foil as a collector of SIBs can greatly reduce the initial price of the battery and its initial weight. In terms of large-scale energy storage, the stable discharge performance of SIBs makes it easy to manage the depth of discharge [7]. The production of SIBs can follow the existing production processes and equipment associated with LIBs [8]. Based on the above-recognized advantages, SIBs have maintained high research interest in recent years. There have been many reports on electrode and electrolyte material systems of SIBs. So far, the energy density of SIBs can exceed 100 W h/kg [9]. Moreover, the cost of electricity can reach less than 0.1/USD (W h) using SIBs [10].

However, reported SIBs are mostly tested at room temperature, and SIBs that are able to work at low temperatures have not received much attention. With the development of SIBs, low-temperature performance is a factor that researchers have to consider. As reported, one of the factors limiting the large-scale application of SIBs is its poor performance at low temperatures [11]. The main reason for this is that sodium ions in batteries move slowly at low temperatures, curbing the voltage variation and reducing the capacity of the battery. Secondly, the reaction of sodium is prone to irreversible phase transition [12]. At the same time, the reaction kinetics are slow at low temperatures [13]. These factors endow the SIBs with greater safety risks and unstable performance in practical applications [14]. Developing materials, electrolyte systems, and processes suitable for sodium-ion batteries is a major challenge facing the development of sodium-ion batteries and deserves attention. This review will summarize the research progress of low-temperature SIBs, mainly from the perspective of electrode and electrolyte material systems, as well as the technologies and methods used to improve the low-temperature performance of sodium-ion batteries. These materials are divided into three parts: cathode material, anode material, and electrolyte. Some representative materials and electrolytes at low temperatures are drawn in Figure 1.

## 2. Cathode Materials

### 2.1. Prussian Blue (PB) and Prussian Blue Analogues (PBAs)

PB(Fe[Fe(CN)_6_]_3_·nH_2_O) and PBAs (A_x_M[M′(CN)_6_]_y_·Z_1−y_·nH_2_O, A = Na, K; M, M′ = Fe, Mn, Ni, etc.; 1 < x < 2, 0 < y < 1; Z: vacancies) have the following advantages. First, their high energy density with a small volume changes per cycle and obtains a favorable rate capability in ambient temperatures. Second, their synthetic process is simple, which makes it suitable for large-scale industrial manufacturing [15,16,17]. Yi Cui et al. [18] synthesized a PB cathode material by nucleating Na_2_Fe[Fe(CN)_6_] cubic nanoparticles on carbon nanotubes (CNT) to form a robust composite material (PB/CNT) as shown in Figure 2a. The material was demonstrated to have fast and stable cycling performance at −25 °C. At the same time, PB crystals maintain a small volume change during the Na^+^ insertion and extraction process. In addition, a larger lattice parameter of the perovskite framework of PB reduces the activation energy of sodium-ion diffusion, and the concatenate of CNT in PB nanocrystals maintains favorable electric contact at low temperatures. The excellent performance of PB and the unique structure allowed PB/CNT cathodes to achieve a discharge capacity of 142 mAh/g at a 0.1 C rate (1 C = 100 mA/g in this review), as shown in Figure 2b, as well as an output specific energy density of 408 W h/kg with coulombic efficiency higher than 99.4% at −25 °C. In multi-lap cycles, the system still had superb stability whose specific retention stays at 86% over 1000 cycles, as shown in Figure 2c. Jian-Ming Dai et al. [19] reported a new method which was an electrostatic spray-assisted coprecipitation method (abbreviated as the ES method), used to prepare the Na_2_Ni[Fe(CN)_6_] cathode, which was named as PBNi-ES. The ES method reported in the article significantly reduces the amount of combined water in the PBNi. In the characterization of SEM, it can be clearly seen that the PBNi prepared through the ES method has more pores and smaller particle size. This new structure has better stability under multiple cycles and a high current density, which also directly enhances the transmission efficiency of Na^+^ and e^−^. In Figure 2d, much higher voltage polarization can be seen. However, after cycling 440 cycles, PBNi-ES exhibited an 87% capacity retention at 0 °C and an 84% capacity retention at −25 °C at a 1 C rate, as shown in Figure 2e. In the work of Yinzhu Jiang et al. [20], Na_1.71_Mn[Fe(CN)_6_]_0.94□0.06_·1.66H_2_O was composed through a facile in situ polymerization method. Combined with 3, 4-ethylenedioxythiophene, (NH_4_)_2_S_2_O_8_ and deionized water, the as-synthesized powder was named as MnHCF@PEDOT-20. The special prepared method and Mn/Fe in PBA promotes the infiltration of these two elements. It further decreases phase change in the cathode, enhancing the cell’s low-temperature performance. It is also worth mentioning that at a low temperature of −10 °C, NaPF_6_ was more stable in the electrolyte, while the degradation of transition metals was reduced. Hence, after a long run of 500 cycles, it still had a remarkable specific capacity of about 70 mAh/g, i.e., nearly 83% compared with its initial capacity, and it had an obvious voltage stage at 3.4 V. Recently, Jerry Barker et al. [21] produced Novasis Prussian Blue material Na_x_MnFe(CN)_6_ on a large scale. With hard carbon as the counter electrode, the full cell exhibited high voltages of 3.25 V at 0 °C, 3.1 V at −10 °C, and 2.9 V at −20 °C, with specific capacities of 90 mAh/g at 0 °C, 88 mAh/g at −10 °C, and 85 mAh/g at −20 °C at a rate of 1 C, as shown in Figure 2f. 

### 2.2. Layered Oxides

In latest research, layered oxides, including Na_x_TMO_2_ (x ranges from 0 to 1; TM is a transition metal), are being increasingly noticed by researchers because they have superior electrochemical performance. These layered oxides also are environmentally friendly and easy to prepare. In Na_x_TMO_2_, there are different Na^+^ conditions and oxide layering ways, and scientists summarized them and divided them into two categories (P2 and O3 types). For now, the main obstacle of layered oxide application in SIB cathodes is that a phase transition is required when charge and discharge take place. Some of them have unfavorable electrochemical behaviors when the temperature goes down, leading to potential safety hazards. Therefore, the use of transition metal oxide cathode materials at low temperatures is more challenging.

In the study of Yufeng Zhao et al. [22], researchers coated a NaTi_2_(PO_4_)_3_ nano-shell on the surface of P2-type manganese-based layered oxide Na_0.67_Co_0.2_Mn_0.8_O_2_ (NCM) to improve its electrochemical performance. Compared with uncoated NCM, the NCM@NTP (7 wt%) sample significantly improved the kinetics of Na^+^ migration and the structural stability of NCM. Therefore, when combined with the sodium metal anode, at −20 °C, the NCM@NTP7 sample exhibited discharge capacities of 122.7 mAh/g at 0.2 C, 110.8 mAh/g at 0.5 C, 104.8 mAh/g at 1 C, 72.7 mAh/g at 5 Cm and 56.1 mAh/g at 10 C. It obtained a capacity retention of 92.3% after 100 cycles at 0.2 C. O3-type oxide is another layered structure cathode for SIBs. Changzhou Yuan et al. [23,24] fabricated 1D ultralong NaCrO_2_ nanowires with a continuous and interconnected framework, which guaranteed large sur-/interfaces for the contacting of electrolyte and active material to provide convenient electronic/ionic migration pathways for rapid charge transfer. Moreover, 1D NaCrO_2_ shows remarkable performance compared with traditional NaCrO_2_ bulk cathodes at low temperatures. When coupled with a hard carbon anode, it exhibited a specific capacity of 108 mAh/g and a stable voltage stage of 3.1 V at a current density of 0.2 C under −15 °C. Even at super high rate of 15 C, its voltage and capacity stayed almost unchanged. Furthermore, 1D NaCrO_2_ obtained a 78.3% specific capacity retention over long charge–discharge life of 100 cycles. It exhibited superior stability than that of bulk NaCrO_2_ cathodes (only a 16.7% specific capacity retention at −15 °C) [25]. Yang-Kook Sun et al. [26] designed a hierarchical columnar structure by assembling Na(Ni_0.75_Co_0.02_Mn_0.23_)O_2_ with another proportion Na(Ni_0.58_Co_0.06_Mn_0.36_)O_2_. This layered structure effectively avoids unnecessary side reactions of the materials with the electrolyte because it expands surface contact proportion by adding more pores. Therefore, this system combined with a hard carbon anode enclosed outstanding electrochemical performance at low temperatures with the rate of 0.5 C through 100 cycles, which delivered specific capacities of 128.8 and 114 mAh/g at 0 °C and capacity retentions of 89% and 92% and −20 °C, respectively. Hou, Yanglong et al. [27] proposed P2-type Na_0.67_Ni_0.1_Co_0.1_Mn_0.8_O_2_ material prepared through reasonable structure modulation. The material offers an excellent Na^+^ conduction rate at −20 °C. At this extremely low temperature, combined with the Na counter electrode, it still delivered a specific capacity of 148.1 mAh/g at the 0.2 C rate. Moreover, when assembled as a full cell, it also had an extraordinary energy density at −20 °C.

### 2.3. Polyanionic-Type Cathodes

Polyanionic-type material is another kind of attractive cathode material for SIBs due to its high operating potential, stable structural framework, and superior safety. According to the type of polyanions, polyanionic-type cathodes can be categorized into six types: phosphates, fluorophosphates, pyrophosphates, mixed phosphates, sulfates, and silicates. So far, mainly phosphates and fluorophosphates are reported for low-temperature SIBs.

#### 2.3.1. Phosphates

In a recent study, Chungang Wang et al. [28] coated a nanoparticle NaFePO_4_ (NFP) with a carbon shell and produced a NFP@C composite, as shown in Figure 3a. The NFP in NFP@C effectively helps to establish ion–electron transfer pathway. Carbon coating enhances ion conductivity; thus, NFP@C obtained ideal electrochemical performance below ambient temperatures. Combined with the hard carbon anode, the full cell delivered a specific capacity of 115 mAh/g at −10 °C and 100 mAh/g at −20 °C. Even after 200 cycles at 0.5 C, 85.5% and 75.8% capacity retentions were obtained with a coulombic efficiency around 100%. Furthermore, when the rates increased to 2 C, the full cell still exhibited a favorable performance at −10 °C and −20 °C. In such low temperatures, the capacity retentions could still reach 87.0% and 75.8%, respectively, when compared with specific retentions at ambient temperatures, as shown in Figure 3b.

Na_3_VCr (PO_4_)_3_ (NVCP) was proven to have better electrochemical performance at low temperatures of −15 °C than that at ambient temperatures of 30 °C. Due to V^3+^/V^4+^ and V^4+^/V^5+^ redox couples in the structure, 1.5-electron cell can be transported during the redox reaction. In 2016, Yong Yang’s group [29] reported that the NVCP three-electrode cell can obtain a high voltage of 3.4 V and an outstanding specific capacity of 93 mAh/g at 0.1 C under −15 °C, as shown in Figure 3c. Later in 2020, the same group [30] further investigated the migration of V in the polyanionic cathode during the Na^+^ storage process by using various in situ/ex situ characterization tools. After a comprehensive analysis, they challenged the traditional common view of the stable framework in this system and proposed that the V migration was associated with the irreversible long-range structural transformation and capacity decay of polyanion-based cathode materials, as shown in Figure 3d. They further demonstrated that V migration could be effectively inhibited at low temperatures (−15 °C) and restored at room temperature via a low-voltage discharge (<1.7 V). 

Shanqing Zhang et al. [31] proposed nanocomposite NASICON-structured Na_3_V_2_(PO_4_)_3_ with low-cost organic carbon derived from sucrose which was abbreviated as NVP@C. They introduced a 3D Na^+^ transportation system in NASICON which maintained a slight voltage fluctuation of 170 mV. When the temperature changed from 23 °C to −10 °C, a specific retention of NVP@C could reach 108 mAh/g, as shown in Figure 3e. Simultaneously, after 500 cycles at a high rate of 10 C at an extremely low temperature of −20 °C, a high capacity retention of 75.8% could be obtained, which was almost the same with that at room temperature. Yunhai Wang et al. [32] reported the Na_3_V_2_(PO_4_)_3_/CNT composite as a cathode for SIBs. In their work, NVP was cross-linked by CNTs, and a full SIB was fabricated with the Na_3_V_2_(PO_4_)_3_/CNT cathode, the Bi anode, and the NaPF_6_-diglyme electrolyte. Such fabrication enabled the full cell to obtain satisfactory cycling stability in the temperature from −15 °C to 45 °C. In addition, Kazuhiko Matsumoto et al. [33] prepared a carbon-coated NASICON-type Na_3_V_2_(PO_4_)_3_ through a sol–gel method and investigated it as a cathode material in ionic liquids. Because the cathode has merits of traditional NASICON-type material, which corner-share the arrangement of the polyhedral units, promoted by carbon coating, it is manifested as an ideal performance at low temperatures. 

In another paper, Yan Wang et al. [34] synthesized the NASICON-structured Na_4_MnCr(PO_4_)_3_ cathode through sol–gel-assisted solid-based way. Then, the system was tested with the Na counter electrode. This half-cell delivered a favorable specific capacity of 100 mAh/g and a high charge–discharge stage around 4.0 V at a rate of 0.1 C under −10 °C. Although these above cathode materials showed relatively high operating voltage under low temperatures, the use of toxic and expensive V and Cr elements remains a critical issue in real applications. The Fe-based NASICON-type cathode is more environmentally friendly. Yongyao Xia et al. [35] prepared a flaky porous Na_3_Fe_2_(PO_4_)_3_ cathode via a spraying and drying method. In the microscopic characterization stage, they found that formed [Fe_2_(PO_4_)_3_] was shaped as cylindrical lanterns. The cathode manifested as a NASICON type underwent two main processes when charge–discharge begun. In the first Na^+^ transfer reaction, there is a phase changing process which is a transportation from original Na_3_Fe_2_(PO_4_)_3_ to Na_4_Fe_2_(PO_4_)_3_. Then, as Na^+^ keeps transferring, the Na_4_Fe_2_(PO_4_)_3_ in cathode turned into Na_5_Fe_2_(PO_4_)_3_. Assembled with a hard carbon anode, the full cell delivered an ideal specific capacity of 74.6 mAh/g in 0 °C and 40 mAh/g at −20 °C with rate of 1 C, as shown in Figure 3f. Meanwhile, MNVP@C nano tubes were designed and synthesized by Changzhou Yuan et al. [36]. They first processed the raw material using a large-scale mechanical stirring method and then annealed it to obtain the target cathode. When testing electrochemical performance, covered with a hard carbon anode, it delivers a specific capacity of 111.3 mAh/g at ambient temperatures, 105 mAh/g at 0 °C, and 95 mAh/g at −15 °C at a 1 C rate. Meanwhile, it exhibited a favorable capacity retention of 91% after 300 cycles at extremely low temperatures of −25 °C. Moreover, Liu, Haimei et al. [37] proposed a Na_4_Fe_3_(PO_4_)_2_P_2_O_7_ structure combined with Mn^2+^ and modified with graphene as an SIB cathode. Both Mn^2+^ doping and graphene modifying effectively enhanced the ion transition; thus, the system showed impressive low-temperature performance. At a rate of 0.2 C at −20 °C, a specific capacity of 85 mAh/g can be obtained. Furthermore, at 0.5 C, its capacity retention is still 96.8% after 180 cycles at −20 °C, indicating that it has more favorable stability in extreme conditions compared with Na_4_Fe_3_(PO_4_)_2_P_2_O_7_ without Mn^2+^ doping.

**Figure 3 nanomaterials-12-03529-f003:**
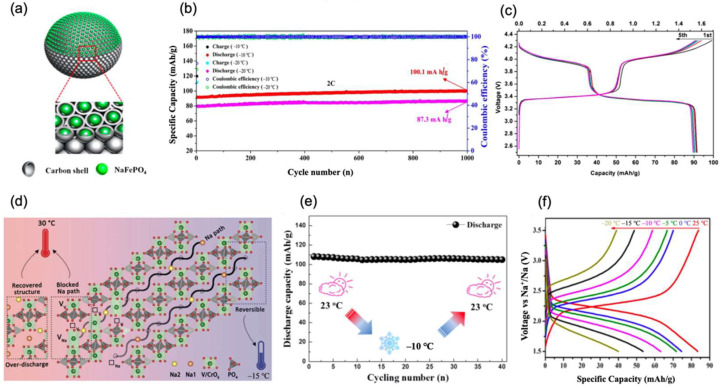
(**a**) The structure of the NFP@C. (**b**) NFP@C cathode charge–discharge in long-term cycling of 1000 cycles at a high rate of 2 C at −10 °C and −20 °C. Reproduced with permission from Ref. [28]. Copyright 2020 Elsevier. (**c**) Electrochemical performance of three-electrode cell at 0.1 C and −15 °C. Reproduced with permission from Ref. [29]. Copyright 2017 American Chemical Society. (**d**) Schematic illustration of Na_2-_xVCP morphological changes in different temperatures. Reproduced with permission from Ref. [30]. Copyright 2020 Wiley-VCH. (**e**) Discharge capacity when the temperature changes between −10 °C and 23 °C with 1 C. Reproduced with permission from Ref. [31]. Copyright 2016 Elsevier. (**f**) The charge–discharge profiles of NFP||Na_x+y_C full cell at 1 C in a range of temperatures. Reproduced with permission from Ref. [35]. Copyright 2019 American Chemical Society.

#### 2.3.2. Fluorophosphates

In the structure of fluorophosphates, highly electronegative F can improve the output voltage. Na_3_V_2_(PO_4_)_2_F_3_ has a special polyanionic structure that consists of a 3D framework with V_2_O_8_F_3_ bi-octahedra connected by PO_4_ tetrahedra and large tunnels where Na^+^ is mobile upon extraction–insertion reactions. Laurence Croguennec et al. [38] produced carbon-coated Na_3_V_2_(PO_4_)_2_F_3_. Additionally, changes in the phase diagram upon cycling were observed by operando X-ray diffraction and other methods, as shown in Figure 4a. When coupled with hard carbon, the full cell showed a high voltage of 3.75 V and a specific capacity of 105 mAh/g at 0.1 C under 0 °C. Meanwhile, a temperature-controlled operando cell was used to determine the phase diagram at 0 °C, which turned out to be mostly unchanged compared to that recorded at 25 °C. Xinglong Wu et al. [39] proposed a high-voltage polyanionic cathode Na_3_V_2_(PO_4_)_2_O_2_F nano-tetraprisms, as shown in Figure 4b, which shows low strain (2.56% volumetric variation) and superior Na transport kinetics in Na intercalation–extraction processes. When assembled with Sb-carbon nanotubes, the full cell exhibited practicable specific capacity of 115 mAh/g and 102 mAh/g at 0.2 C under −5 °C and −15 °C, respectively. Even at −25 °C, its capacity reached 96 mAh/g (about 76.4% capacity retention compared to that at 25 °C), implying the superior low-temperature kinetics. 

Maowen Xu et al. [40] hybridized Na_3_V_2_O_2_(PO_4_)_2_F with reduced graphene oxide to enhance its electronic conductivity. It is proved that the open framework is more conductive to sodium-ion migration during charging–discharging processes at low temperatures. Hence, when combined with the polymer electrolyte, the Na_3_V_2_O_2_(PO_4_)_2_F/Na half-cell demonstrated specific capacities of 107 mAh/g, 84 mAh/g, 74 mAh/g, and 58 mAh/g, even under temperatures of −5 °C, −15 °C, −20 °C, and −25 °C at 1 C, respectively. After cycling at 1 C for 190 cycles under −25 °C, it still has a capacity retention of 99.6%, demonstrating superior low-temperature stability, as shown in Figure 4c. One mixed phosphate is reported for low-temperature SIBs. Shu-Lei Chou et al. [41] synthesized tunable Na_4_Fe_3_(PO_4_)_2_(P_2_O_7_)/C nanocomposite via a facile one-step sol–gel method with high phase purity and uniform carbon coating, owing to the 3D sodium diffusion pathways and high sodium diffusion coefficient in Na_4_Fe_3_(PO_4_)_2_(P_2_O_7_), as shown in Figure 4d. The composite showed 84.7 mAh/g at 0.2 C, −20 °C, and was maintained over a 92% capacity retention after 250 cycles, as shown in Figure 4e.

Herein, some representative cathode materials were selected, and Figure 5 was plotted according to voltage and specific capacity. Since the current density only has a slight difference, we can roughly draw the following conclusion. It is clearly concluded from Figure 5 that the specific capacity of most cathode materials can be maintained at 100 mAh/g at about −20 °C. The electrical property of the battery is slightly influenced by low temperatures, at which the voltage is maintained at about 3 V. Among these materials, Prussian blue is the most used low-temperature cathode material. It is often reported that it can be produced at large scales due to its ideal stability. Some cathode materials, such as sodium iron phosphate, have high electrical properties but low voltages. For one cathode material, different electrolytes also cause great differences in electrochemical performance at low temperatures. For example, although carbon-coated Na_3_V_2_(PO_4_)_3_ is used as a cathode material, as shown in Figure 5, the conventional organic electrolyte used previously and the ionic liquid electrolyte used in the later results have different electrochemical capacities.

In the end of the cathode materials’ section, it is worth mentioning that PB and PBAs are also our significant options. The unique perovskite skeleton in this cathode material has a large lattice parameter, which facilitates the diffusion of Na^+^. At the same time, CN^−^ increases the opening of the perovskite surface. Meanwhile, the smaller charge and the three bonds in CN^−^ further reduce the contact between Na^+^ and the anionic *p*-π electrons through the passing site face. In following experiments, the introduction of carbon nanotubes further enhanced the conduction of Na^+^; hence, the cathode has superior electrochemical performance at low temperatures. As mentioned above, the cathode materials obtained by introducing structural adjustment methods or through special preparation methods in the work of other research groups also exhibited outstanding low-temperature performance. Furthermore, PB and PBAs, which originated with the group of Professor John B. Goodenough of the University of Texas at Austin in 2010, were used in SIBs as part of large-scale commercialization attempts produced by Novasis Energies, Inc. (Novasis). This cathode still has stable cycling performance at low temperatures, while other SIB cathode materials have rarely been reported for large-scale commercial applications. We believe that PB and PBAs are the most ideal cathode materials in the future. 

## 3. Anodes

### 3.1. Hard Carbon

As for SIB anode materials, hard carbon (HC) is the most practicable. HC is usually pyrolyzed by a high-molecular-weight polymer, so these carbons are difficult to graphitize at high temperatures above 2800 °C. Common HC includes resins such as phenolic resins, polyalcohols, epoxy resins, organic polymers, pyrolysis carbons, and carbon blacks (such as acetylene black). Due to the large interlayer spacing of hard carbon, it is conducive to the diffusion of ions between carbon layers and is also conducive to rapid charge and discharge. HC has a high specific capacity and is a promising carbon anode material with many current applications and research studies. However, at low temperatures, its kinetics slow down, resulting in serious polarization inside the battery. The aggravated polarization further reduces the electrochemical plateau to around 0 V. Due to the decrease in voltage, the deblocking ability of sodium ions to graphite anodes is greatly weakened, leading to an unfavorable specific capacity. This makes the low potential, which is originally an advantage of anode materials, one of the factors for the deterioration of electrochemical performance. Meanwhile, at lower temperatures, the formation of sodium dendrites is more likely to damage the SEI membrane, thereby reducing its stability and specific charge–discharge capacity. 

Scientists aim to solve these problems through covering carbon and graphite on it. To conquer this bottleneck, M.R. Palacín et al. [42] developed a simpler physical method to replace a traditional chemical method to realize carbon covering, which obtained uniform particles. They then tested a half-cell at 0 °C and −15 °C at a low rate of 0.1 C and compared it with that tested at 25 °C. It is worth mentioning that the specific capacities obtained at low temperatures (260 mAh/g at 0 °C and 265 mAh/g at −15 °C) were close to those obtained at 25 °C (290 mAh/g), as shown in Figure 6a. In another group, Xing-Long Wu et al. [43] applied HC paper composed of carbon nano tubes by using tissue as a precursor to improve the performance of the HC anode, which is shown in Figure 6b. Thanks to the “adsorption-intercalation” mechanism of HC paper in the 1 mol/L NaCF_3_SO_3_/diglyme electrolyte, reactions in the battery were accelerated at low temperatures. At a rate of 0.5 C, the reversible capacity of the HC electrode is about 320 mAh/g at −5 °C, 310 mAh/g at −15 °C, and 302 mAh/g at −25 °C, as shown in Figure 6c. Moreover, another highlight of this system is its excellent low-temperature cycle stability after ultra-long cycle life at low temperatures. At −15 °C, with a high rate of 5 C, the system exhibited an 81% capacity retention after 1000 cycles. 

Moreover, Biao Zhang et al. [44] used longan peel as a precursor for preparing HC to obtain ideal activated carbon modifications, as shown in Figure 6d. This HC anode exhibited outstanding low-temperature performance in the electrochemical determination process when coupled with the Na_3.5_V_2_(PO_4_)_2_F_3_ cathode. A discharge capacity of 400 mAh/g was achieved for the half-cell and 244 mAh/g was achieved for the full cell at a rate of 0.1 C at −20 °C. However, the electrochemical performance of the full cell decayed dramatically at −20 °C because of more serious solid electrolyte interphase layers produced, compared with that in 25 °C after cycling revealed by the spectrum, as shown in Figure 6e. It could return to normal electrochemical function when the temperature returns to an ambient temperature, as shown in Figure 6f. However, it is proved that the polarization in the HC-based full cell is considerably reduced than that in the half-cell, due to the severer side reaction of Na electrodes with electrolytes than that of HC electrodes. In the research of Chen, W. et al. [45], a new molten diffusion–carbonization method to prepare HC has been proposed, which effectively reduced pore diameter from more than 1 nm to less than 0.5 nm. In this way, interfacial contact between the electrolyte and pores could be reduced. Therefore, the designed carbon anode exhibits a real capacity of 5.32 mAh/cm^2^ at a low temperature of −20 °C. Moreover, its capacity retention at −20 °C is around 87% compared to that obtained at ambient temperatures. In summary, because of the favorable electrochemical performance of HC below 0 °C, it can be constantly used to couple with the target cathode to study the low-temperature performance of full cells, which also proved the practicality of the HC anode.

### 3.2. Amorphous Selenium and Metal Selenides

Se materials have also attracted extensive attention as a non-toxic, cheap, and abundant material. Amorphous selenium (a-Se), as a semiconductor, has marvelous electrical conductivity and a comparable high volumetric capacity, which make it attractive for anode material in SIBs. A-Se is also an abundant, environmentally friendly, and chemically stable material. Compared with carbon-based materials which is difficult to form intercalation compounds in Na-ion batteries, the a-Se alloy compounds have excellent electrochemical performance. Transition metal selenides such as MoSe2, FeSe_2_, CoSe_2_, and CuSe have received widespread attention for their low reaction energy consumption and superior theoretical specific capacity. However, when the temperature is reduced, during the sodium/disodium process, the conductivity of electrons and ions is low and the volume expansion is severe, resulting in electrode chalking and a loss of electrical contact, leading to extremely low cycle stability. Researchers improved their low-temperature performance by designing the structure of the unique Se anode material or modifying it with carbon material to accelerate the transfer of electrons and ions during charge and discharge, while increasing the active site. 

Xing-Long Wu et al. [46] coated a-Se on a three-dimension network structure with reduced graphene oxide nanolayers to produce the Se/graphene (3DSG) anode. The three-dimension network based on graphene oxide nanolayers in 3DSG, as shown in Figure 7a, provided ideal transport paths for Na^+^. When cycled in half-cells with the Na counter electrode, it exhibited favorable electrochemical performance when the temperature went down. For example, it showed a reversable specific capacity of 250 mAh/g at −5 °C and 180 mAh/g at −15 °C at a rate of 2 C. Moreover, it still had satisfactory capacity retentions of 96.2% and 98% after 1000 cycles, as shown in Figure 7b. Luo Wen-Bin et al. [47] prepared NbSSe nanosheets through calcination as the SIB anode material. The two-dimensional nanosheets stabilizes interlayer band gap and improves electronic transformation. One prominent advantage of this system is the combination of two different kinds of anionic ligand characteristics which improved electrochemical performance and stability at low temperatures. At a low temperature of 0 °C, it showed a specific capacity of 136 mAh/g. Furthermore, after a long life span of 500 cycles at 0 °C, it exhibited a satisfactory specific retention of 92.67% at a rate of 0.2 C.

Compared with amorphous selenium, metal selenides own higher electrical conductivity; thus, they are more favorable in low-temperature SIBs. From the SEM image in Figure 7c,d, it can be easily concluded that selenide carbon composite has a one-dimensional structure which represents considerable superficial area, resulting in an ideal Na^+^ diffusion efficient. Moreover, as a widely acknowledged semiconductor, SnSe is abundant and environmentally friendly. In addition, it has a high theoretical capacity of 780 mAh/g for sodium-ion storage. In the research of Ming Zhang et al. [48], a SnSe@carbon nanofiber (SnSe@CNF) was designed by adding selenium powder into a tin solution. After annealing and covering with a carbon nanofiber, SnSe@CNF was formed which improved the traditional method using a thermal method and its nanofiber structure, as shown in Figure 7e. The half-cell showed a stable capacity of 267 mAh/g at 0 °C after 100 cycles at a 1 C rate. In another report, Chunming Zheng et al. [49] also chose carbon nanotubes to integrate ZnSe. Through a unique hydrothermal method, ZnSe was perfectly combined with carbon nanofibers to form the anode material and provide a network for Na^+^ diffusion. In addition to the merits of carbon nanotubes mentioned above, the nanofiber ZnSe structure can also play a supporting role because of its superior bearing ability. It can stretch when the pressure is reduced and prevent the structure from collapsing when the pressure increases, resulting in improved electrochemical performance at low temperatures. For instance, the half-cell delivered a discharge capacity of 267.0 mAh/g at a rate of 1 C. Moreover, when it came to 600 cycles, an 83.3% capacity was still retained, demonstrating better cycling performance than pure ZnSe at −10 °C, as shown in Figure 7f.

**Figure 7 nanomaterials-12-03529-f007:**
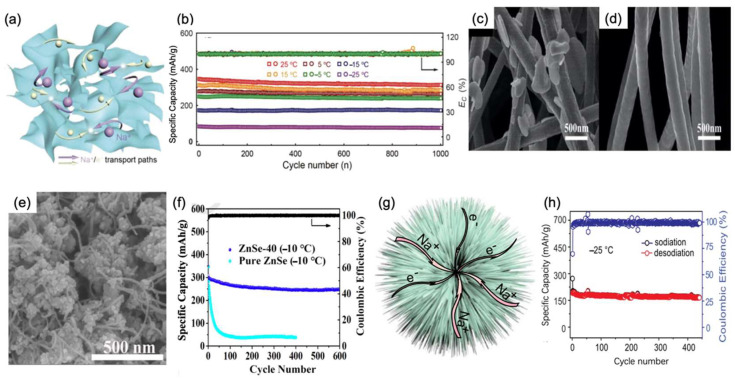
(**a**) Three-dimensional framework of 3DSG, showing the transportation of Na^+^ and e^−^. (**b**) Long-term cycling in a range of temperatures at a 2 C rate. Reproduced with permission from Ref. [46]. Copyright 2018 WILEY-VCH. (**c**,**d**) SnSe@CNFs optical image without the addition of Se powder. Reproduced with permission from Ref. [48]. Copyright Springer Nature 2019. (**e**) SEM image of ZnSe-40. (**f**) Long-term cycling performance of ZnSe-40 and pure ZnSe at −10 °C. (**g**) Schematic illustration of Fe_7_Se_8_@C 3D structure with Na^+^ and e^−^ transportation routes. (**h**) Specific capacity and coulombic efficiency of cl-Fe_7_Se_8_@C//NVPOF full cell after 400 cycles at −25 °C. Reproduced with permission from Ref. [50]. Copyright 2019 American Chemical Society.

In another work, Jing-Ping Zhang et al. [50] constructed a coral-like cl-Fe_7_Se_8_@C structure with Fe and Se material. One of the advantages of this system is that one-dimensional carbon nanotubes in the three-dimensional structure facilitates the transfer of zero-dimensional Fe_7_Se_8_ nanospheres, as shown in Figure 7g, resulting in favorable performance. To former test its low-temperature stability, the cl-Fe_7_Se_8_@C anode was assembled with a Na_3_V_2_(PO_4_)_2_O_2_F cathode which already showed an ideal capacity when the temperature decreased. The full cell exhibited a specific capacity of 166 mAh/g after 440 long-term cycles at 0.5 C. This means that its capacity fading per cycle was reduced to about 0.15% at an extremely low temperature of −25 °C, as shown in Figure 7h. In another study, FeSe_2_/rGO hybrids with a 3D hierarchical structure, prepared in a traditional hydrothermal way, were developed by Ye, Zhizhen et al. [51]. Not only did the unique anode show impressive performance at ambient temperatures, but it also exhibited ideal electrochemical performance and stability at low temperatures. At low temperatures of 0 °C, −10 °C, and −20 °C when they were compared with room temperature, its capacity retentions were still 94%, 90%, and 85%, respectively, indicating remarkable stability. It is worth mentioning that even at an extremely low temperature of −30 °C, its specific capacities could still reach 349 mAh/g, which is 68% compared with that at room temperature. Meanwhile, at −40 °C, it reached 278 mAh/g, which is 53% compared with that at room temperature at a 10 C rate. In addition, after 200 cycles, its specific capacity remained 217 mAh/g at −40 °C.

### 3.3. NaTi_2_(PO_4_)_3_

Low electrical conductivity and the ion diffusion coefficient affect the multiplicity performance of the NaTi_2_(PO_4_)_3_ electrode, hindering their application in electric vehicles. NaTi_2_(PO_4_)_3_ (NTP), with a sodium superionic conductor (NASICON) structure, has large ion channels for the insertion and extraction of sodium ions at a faster rate, thereby obtaining high electrical conductivity. At the same time, NTP can also be made into a one-dimensional structure, which is rich in sodium insertion sites. NTP also has the advantages of high capacity and stable structure, which gives it excellent electrochemical performance at low temperatures. To enhance the multiplicity performance of NTP, carbon coating is more adopted by scientists since it not only effectively promotes the conductivity of it, but also reduces particle size and prevents the oxidation of metal ions. Chunhua Chen et al. [52] mixed traditional NTP with Na_3_V_2_(PO_4_)_3_ (NVP). Then, he coated the mixture with graphene-like layers to synthesize the new nano-porous electrode (NTP@C-2), as shown in Figure 8a,b. It is proved that NTP@C-2 had more sp2-type carbon than carbon-coated NTP without NVP, resulting in more favorable electrochemical performance. At a rate of 0.2 C, NTP@C-2 exhibited a charge capacity of 105 mAh/g at 0 °C. At −10 °C, its specific charge capacity was demonstrated to be 108 mAh/g. At an extremely low temperature of −20 °C, NTP@C-2 showed specific capacities of 102 mAh/g, 98 mAh/g, and 61.1 mAh/g at a rate of 0.2 C, 10 C, and 20 C, respectively. Meanwhile, in Figure 8c, it is obvious that, compared with other two similar materials, NTP@C-2 delivered more stable electrochemical performance, both in low temperatures and ambient temperatures. 

In addition, Ruben-Simon Kühnel et al. [53] designed the full cell with an NTP anode, a polyanionic cathode Na_3_(VOPO_4_)_2_F, and an ionic liquid electrolyte, which manifested as an ideal low-temperature electrochemical performance. It will be discussed in the ionic electrolyte section. Zhanliang Tao et al. [54] coated NTP with carbon which boosts the transportation of Na^+^ and brings about fast reaction kinetics of the cell. Moreover, when assembled with the Ni(OH)_2_ cathode and the 2 M NaClO_4_ electrolyte, the full cell demonstrates remarkable low-temperature rate performance through a dual-ion reaction. When at the high of 10 C, after 500 cycles, the capacity retention showed an unusual perfect capacity retention of 100% at −20 °C which was much higher than that of 25 °C. The full cell also delivered a specific capacity of 82 mAh/g at 0 °C, 78 mAh/g at −10 °C, and 70 mAh/g at −20 °C with a voltage stage of 1.25 V, as shown in Figure 8d. Furthermore, even after ultra-long cycles at a rate of 10 C, the full cell had a remarkable capacity retention of 85% at −20 °C, as shown in Figure 8e.

**Figure 8 nanomaterials-12-03529-f008:**
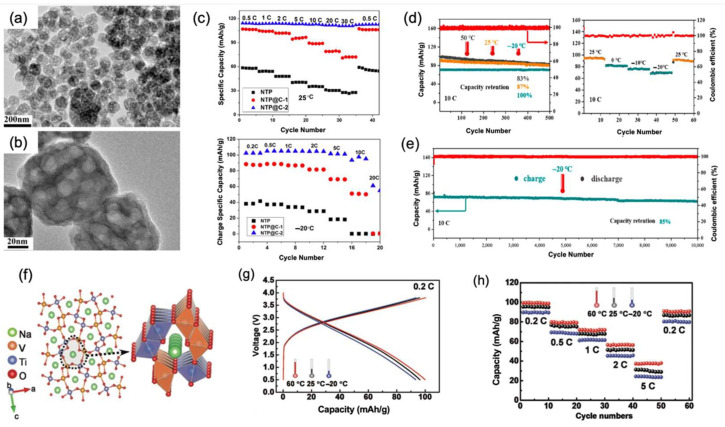
(**a**) Optical image of NTP@C-2 in 200 nm. (**b**) Optical image of NTP@C-2 in 20 nm. (**c**) Electrochemical performance of NTP (black), NTP@C-1 (red), and NTP@C-2 (blue). The picture above represents these three anodes at 25 °C, while the picture below represents these three anodes at −20 °C. Reproduced with permission from Ref. [52]. Copyright American Chemical Society. (**d**) In the figure on the left, the blue line represents the full cell’s electrochemical performance tested at −20 °C, the yellow line represents it tested at 25 °C, and the black line represents it tested at 50 °C with a high rate of 10 C. The figure on the right shows it cycles in different temperatures at 10 C. (**e**) Full cell tested for ultra-long 10,000 cycles at −20 °C with a high rate of 10 C. Reproduced with permission from Ref. [54]. Copyright 2019 American Chemical Society. (**f**) Schematic illustration of microstructure of NaV_1.25_Ti_0.75_O_4_. (**g**) Electrochemical performance of the NaV_1.25_Ti_0.75_O_4_/Na_0.8_-Ni_0.4_Ti_0.6_O_2_ full cell. The red, black, and blue lines represent the full cell tested at 60 °C, 25 °C, and −20 °C with rate of 0.2 C, respectively. (**h**) The stability of the full cell was tested when the rate changed in a range from 0.2 C to 5 C at temperatures of 60 °C, 25 °C, and −20 °C. Reproduced with permission from Ref. [55]. Copyright 2018 WILEY-VCH.

Besides the above-mentioned three types of anodes, high-crystallinity Ti-based oxide NaV_1.25_Ti_0.75_O_4_ was also prepared and investigated in a high-crystalline structure as an all-climate anode material [55]. NaV_1.25_Ti_0.75_O_4_ has typical 1D channels of post-spinel, wherein the channel skeletons are constituted by VO_6_ and TiO_6_ octahedrons in the form of vertices or edge-shared linking, as shown in Figure 8f. Attributed to the robust framework and speedy expressway for sodium uptake and release in post-spinel NaV_1.25_Ti_0.75_O_4_, this material can achieve a high D_Na_^+^ of 3.48 10^−11^ cm^2^ s^−1^, even when the temperature dropped to −25 °C. Hence, assembled with the Na_0.8_Ni_0.4_Ti_0.6_O_2_ cathode, the full cell’s discharge capacity can reach 93 mAh/g at a 0.2 C rate, as shown in Figure 8g. Moreover, after 200 long-term cycles, an 84% capacity retention was maintained and a 7% capacity variation was detected at a high rate of 5 C at −20 °C compared with that at 25 °C, as shown in Figure 8h. It is also worth mentioning that even after cycling at different rates at −20 °C, it can return to the original performance after 60 cycles.

In summary, representative anode materials were selected and plotted according to voltage and electrical properties, as shown in Figure 9. It can be seen from Figure 4 that the anode materials have higher electrical properties at low temperatures than the cathode materials. Most of the anode materials have a specific capacity of about 350 mAh/g and a voltage stage of about 0.5 V. Since the current density only has a slight difference, we can roughly draw the following conclusion. Carbon-based materials or structurally modified C materials have the best electrochemical performance and stability in these materials. Among them, the most used HC shows the most excellent performance at low temperatures. In addition, the electrode materials formed by combining Se and V nonmetal elements with metal elements have high electrical properties. The titanium-containing cathode has high stability in the literature but shows relatively a low specific capacity. In conclusion, other cathode materials should be explored based on their capacity at the same time as widely applying carbon-based materials, and the application of SIBs at low temperatures should be comprehensively studied by combining the cathode materials and the electrolyte.

Among all the above-mentioned anode materials, HC anode materials are most valued due to their low potential versus Na^+^/Na and high cycling stability during de/sodiumization. The smooth electrochemical plateau of HC is attributed to the intercalation and deintercalation of Na^+^ into graphite interlayer gap which is larger than 0.37 nm. HC, which has been widely used as a node material, also has higher stability and better electrochemical performance than other anode materials at low temperatures. The most important thing is that the precursor of HC is cheap and environmentally friendly, which has incomparable practical advantages compared to other anode materials. General HC precursor materials are composed of various biopolymers, including hemicellulose, cellulose, lignin, pectin, protein, free sugars, etc. In our article, both longan peel and tissue paper used as precursor materials showed satisfactory low-temperature performance. The main progress now lies in annealing the precursor and further improving the porosity inside the HC to increase the Na^+^ diffusion rate and improve its electrochemical performance at low temperatures.

## 4. Low-Temperature SIB Electrolytes

As an important component of SIBs, the electrolyte is a medium for ion conduction that provides pathway to support electrochemical reactions on electrodes [56]. The electrolyte not only affects the migration rate of Na^+^, but also participates in the formation of the SEI [57]. Therefore, the electrolyte is a key factor determining the electrochemical performance of SIBs. The unfavorable electrochemical character of electrolytes at low temperatures is also a considerable bottleneck to the application of SIBs [58]. Low-temperature conditions seriously affect the performance of the electrolyte. Firstly, the viscosity of the electrolyte increases, which not only leads to a decrease in ion migration rate and conductivity [59], but also causes the poor wettability of the electrode and the membrane [60]. In addition, because the resistance of SEI between the electrode and the electrolyte increases, the electrolyte becomes less compatible with electrode materials at low temperatures [61]. All these unfavorable factors damage the energy density and cycle stability of the battery.

Therefore, electrolytes that can be applied under low-temperature conditions should have the following characteristics: (a) favorable stability; (b) good ionic conductivity; (c) a wide electrochemical window; (d) no reaction with other battery components; (e) high safety and low toxicity, and (f) can meet the cost requirements for practical applications. The above characteristics of electrolytes essentially depend on the property and selection of the electrolyte salt [62]. While a limited number of efforts are being directed to the search for new electrode materials for SIBs at low temperatures, studies dealing with the electrolytes are much scarcer. Herein, we summarize recent research on SIBs at low temperatures and divide these electrolytes into four sections according to their composition: organic electrolytes, aqueous electrolytes, ionic liquid electrolytes, and solid/quasi-solid electrolytes. 

### 4.1. Organic Electrolytes

Organic electrolytes are usually a mixture of sodium salts (ca. NaClO_4_, NaPF_6_) with different carbonic acid derivative (ca. PC = polycarbonate, EC = ethylene carbonate, DMC = dimethyl carbonate, DEC = diethyl carbonate, EMC = ethyl methyl carbonate, etc.). Meanwhile, the performance of electrochemical SIBs can be significantly improved by changing the composition and ratio of carbonic acid derivatives or slightly adding some other additives. At the same time, matched with suitable anode and cathode materials, the SIB can be manifested given its favorable electrochemical capacity and stability. Among sodium salts, NaClO_4_ is more abundantly studied at low temperatures. In addition, the PC solvent has a freezing point of −48 °C and an outstanding dielectric constant which effectively avoids any solidification of the electrolyte system and improves the low-temperature performance of SIBs by increasing the ionic conductivity. However, the single-component PC is less compatible with the electrodes, so efforts are devoted to solving the problem with additives [63]. It is proved by Haoshen Zhou et al. [64] that the addition of fluoroethylene carbonate (FEC) hugely improved the PC electrolyte low-temperature performance. The electrolyte of 1 M NaClO_4_ dissolving in PC with the addition of 2 vol% FEC exhibited a specific capacity of 94 mAh/g at a 0.2 C rate with an 8.7% capacity decline compared to that the 25 °C in the half-cell test with a NaV_1.25_Ti_0.75_O_4_ cathode at −20 °C. Combined with the Na_0.8_Ni_0.4_Ti_0.6_O_2_ anode, the Na_0.8_Ni_0.4_Ti_0.6_O_2_//NaV_1.25_Ti_0.75_O_4_ full cell manifested a specific capacity of 93 mAh/g at a 0.2 C rate and maintained a 84% retention capacity over 200 cycles at a 1 C rate. Chungang Wang et al. [28] added 5 vol% EFC in the PC. Cycled at low rate of 0.5 C, the NFPNCs maintained 114.7 mAh/g and 100.6 mAh/g with coulombic efficiency around 100% at −10 °C and −20 °C, respectively. These two kinds of electrolyte were testified by decreasing the polarization in cells to improve the electrochemical function.

Conventional solvent EC has a high dielectric constant and outstanding membrane formation, but its high melting point and viscosity limit the application of SIBs at low temperatures, so EC needs the addition of a suitable co-solvent to improve the low-temperature performance of SIBs [64]. It is commonly used as a cosolvent with the PC in low-temperature SIBs, and good low-temperature results have been achieved. Yi Cui et al. and Shanqing Zhang et al. [18,31] both used 1 M NaClO_4_ dissolved in EC/PC 1:1 (vol) as an electrolyte combined with PB/CNT and NVP@C, before then being tested in a half-cell. The PB/CNT half-cell disclosed 155 mAh/g and 105 mAh/g specific capacities in 0.3 C at 0 °C and −25 °C, respectively. Even at a 2.4 C rate and super long cycles of 1000, the cell could still maintain high capacity retentions of 81% and 86%, respectively. As for the NVP@C, an oblivious decay of the specific capacity can be noticed with the temperature decreasing. As the temperature changed by ranges of 0 °C, −10 °C, and −20 °C, the specific capacity changed from 120 mAh/g to 116 mAh/g at a 0.2 C rate. Meanwhile, at −20 °C, with a rate of 10 C, the retention capacity could still remain 75% after 500 cycles. In the works of XingLong Wu et al. and JingPing Zhang et al. [46,50], the FEC was also used as an additive to the 1 M NaClO_4_ with a EC/PC cosolvent system which magnificently enhanced the specific capacity. The specific capacity of the cl-Fe_7_Se_8_@C half-cell reached 425 mAh/g with a current density of 0.2 A/g, which decreased to 350 mAh/g from 0 °C to −25 °C. Compared with the performance at 2 C at room temperature, the capacity retentions of the cell reached about 96% at 0 °C, 82% at −15 °C, and 68% at −25 °C. Meanwhile, the experiment of 3DSG composite half-cells demonstrated a high specific capacity of 375 mAh/g at −5 °C, 300 mAh/g at −15 °C, and 250 mAh/g at −25 °C, respectively, in a large current density of 2 A/g. Moreover, the retention capacities were 96.2%, 98%, and 90.4%, respectively, over 1000 cycles at 20 C. 

Moreover, cosolvents formed by EC and other carbonic acids also performed stable low-temperature electrochemical functions. EC dissolved with DMC (1:1 vol) after adding 1 M NaClO_4_ was investigated in the low temperatures of 0 °C and −25 °C by Jian-Ming Dai et al. [19]. Assembled with PBNI-ES, the half-cell displayed specific capacities of 79.3 mAh/g and 65.1 mAh/g at a rate of 0.5 C, respectively. They also investigated the factors limiting the electrochemical performance of the cell at low temperatures, and their optical images (as shown in Figure 10a) and EIS figures (as shown in Figure 10b) exhibit electrolyte freezing when the temperature goes down (from left to right in Figure 10a). This change leads to the slow diffusion of Na^+^ and decreases in conductivity, as characterized in Figure 10b, but in combination with the previous discussion of PBNi-ES, it still maintains excellent low-temperature performance. Additionally, in the half-cell of NTP@C-2 with 1 M NaClO_4_ dissolved in the EC/DEC (1:1 vol) cosolvent, it delivers inconspicuous capacity fluctuation as the temperature decreased [52]. In the work of M.R. Palacín et al. [42], at a rate of 0.26 C, the capacities reached 105 mAh/g, 110 mAh/g, and 102 mAh/g in the low temperatures of 0 °C, −10 °C, and −20 °C. 

Furthermore, compared with NaClO_4_ dissolved salts in the electrolyte, NaPF_6_ is manifested as no big difference in electrochemical performance. Meanwhile, it is proven to have an ideal low-temperature performance. In the research of the carbon-covering HC anode, 1 M NaPF_6_ solution in EC and PC mixture (1:1 wt) with an additive of 10 wt% DMC was chosen by M.R. Palacín et al. as the electrolyte. At low temperatures of 0 °C and −15 °C, it delivered favorable specific capacities of 260 mAh/g and 265 mAh/g at a rate of 0.1 C, respectively, which showed a slight degradation compared to that at 25 °C (290 mAh/g). Additionally, Yang-Kook Sun et al. [59] tested a hierarchical ternary polymeric cathode using the EMS: FEC 98:2 (vol) cosolvent in 0.5 M NaPF_6_ solution. Attributing to the low-temperature function of the unique cathode and the favorable electrolyte, the full cell sandwiched by HC anodes demonstrated appreciable specific capacities of 130 mAh/g and 100 mAh/g at a 0.75 C rate with slightly different retention capacities of 89% and 92% after 100 cycles at a 0.75 C rate. In Figure 10c, the conventional organic electrolytes based on sodium salt NaClO_4_ and NaPF_6_ solvents with organic solvents have a wide electrochemical window and ideal thermodynamic stability. Meanwhile, in the group of Huang, Yunhui et al. [65] tested the conventional organic electrolyte in the Na_3_V_2_(PO_4_)_2_O_2_F half-cell. By assembling into Na/ Na_3_V_2_(PO_4_)_2_O_2_F batteries, a high capacity retention of 93.1% was realized after 1000 cycles at 1 C. Under an ultra-high rate of 30 C and a low temperature of −30 °C, the discharge capacities still reached 89.2 and 92.1 mAh/g, respectively. This also proves that SIBs with different electrode materials mentioned above, combined with conventional organic electrolytes, exhibit practical electrochemical capabilities at low temperatures. Therefore, with the easy synthesis processes, the conventional organic electrolyte is the most suitable choice for SIB at low temperatures.

**Figure 10 nanomaterials-12-03529-f010:**
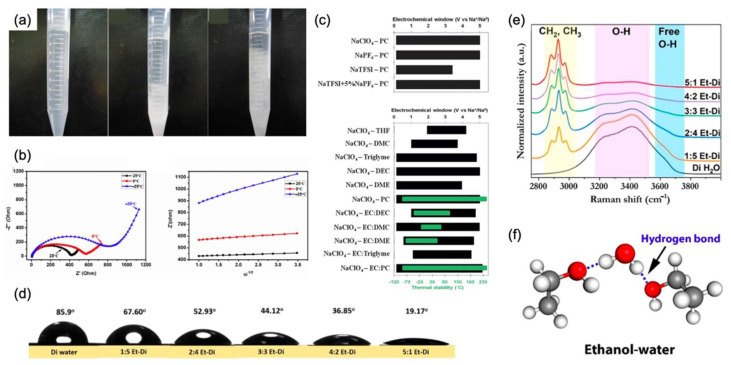
(**a**) On the left is the optical picture of the electrolyte in PBNi-ES at −25 °C. In the middle is the electrolyte at 0 °C. On the right is the electrolyte at 25 °C. (**b**) In the EIS spectra, the black line represents performance tested at 25 °C, the red line represents 0 °C, and the blue line represents −25 °C. Reproduced with permission from Ref. [19]. Copyright 2018 Elsevier Ltd. (**c**) The black bar represents electrochemical windows for different electrolytes, while the green bar represents their thermal stabilities. The Y axis represents solvents based on PC with 1 M sodium salts and 1 M NaClO_4_ with different other solvents. Reproduced with permission from Ref. [59]. Copyright The Royal Society of Chemistry 2012. (**d**) The picture compares different contact angles of various proportions of ethanol–water. (**e**) Raman scattering spectra of various proportion of ethanol–water. (**f**) The picture represents the new type of hydrogen bond in the ethanol–water system. Reproduced with permission from Ref. [66]. Copyright 2020 American Chemical Society.

### 4.2. Ionic Liquid Electrolytes

Ionic liquids are ionic, noncombustible, and nonvolatile, and salt-like electrolytes can provide broad electrochemical window, as well as high stability and safety. Moreover, it has a broad temperature range that gives a promising low-temperature application horizon [67]. Imidazolyl-based ionic liquids, especially those with 1-ethyl-3-methylimidazolyl as the cation, have high conductivity and low viscosity, and have been shown to possess impressive low-temperature properties in recent studies. Kazuhiko Matsumoto et al. [33] reported the electrolyte of Na [bis (fluorosulfonyl) imide]-[1-ethyl-3-methylimidazolyl] [bis(fluorosulfonyl)amide] (NaFSI-EMIFSI) system with various molar fractions. After testing the system with the Na superionic conductor (NASICON)-type NVP electrode, it was shown that the molar fraction of 2:8 (NaFSI: EMIFSI) exhibited the best low-temperature electrochemical performance. The specific capacities were 78.1 mAh/g and 58.6 mAh/g with a 0.1 C rate at −10 °C and −20 °C, respectively. In another work of Giovanni Battista Appetecchi et al. [68], the conductivities of several bis(trifluoromet-hylsulfonyl) imide (TFSI) and FSI with NaTFSI as ionic liquids were reported. It can be easily concluded that EMI-based electrolytes have favorable low-temperature conductivity, as shown in Figure 11d. In this way, NaTFSI-EMIFSI (molar ratio 1:9) and NaTFSI-EMITFSI (molar ratio 1:9) showed remarkable conductivities of 1.1*10^−3^ S/cm and 3.8*10^−4^ S/cm, respectively. 

Meanwhile, pyrrole ionic liquids are cyclic quaternary amine ionic liquids with similar physicochemical and structural properties as chain quaternary amine ionic liquids. In the study of Rika Hagiwara et al. [69], NaFSI-[N-methyl-N-propylpyrro-lidinyl] [FSI] (NaFSI-C_3_C_1_pyrrFSI) was proven to have a favorable electrical conductivity (9.8*10^−4^ S/cm) at a low temperature of 0 °C. When NaFSI-C_3_C_1_pyrrFSI was assembled with NaCrO_2_, about 85% and 51% of the initial discharge capacities were maintained (100 mAh/g and 60 mAh/g) at 0 °C and −10 °C, respectively. It is also worth mentioning that after a series of temperature changes from 90 °C to −20 °C and back to 90 °C, the discharge capacity returned almost completely to its initial value, indicating that no significant degradation occurred at −20 °C, as shown in Figure 11c. Moreover, it also showed outstanding stable performance, of which retention capacity was still 95% after 500 cycles with a current density of 1 C. Meanwhile, when NaFSI-C_3_C_1_pyrrFSI was assembled with Na_2_FeP_2_O_7_, the half-cell demonstrated a specific capacity of 80 mAh/g at 0 °C, 67 mAh/g at −10 °C, and 42 mAh/g at −20 °C [70]. Moreover, after a series of temperature gradient changes from 90 °C to −20 °C and back to 90 °C, the specific capacity exhibited only a slight change. In another study of G.B. Appetecchi et al. [71], they investigated a family of PYR_14_TFSI-NaTFSI (PYR_14_: N-alkyl-N-methylpyrro-lidinyl cations with 14 carbon atoms in the alkyl side chain) mixtures and proved that with a molar fraction of 9:1, the electrolyte showed remarkable conductivity (2.2*10^−3^ S/cm) at −30 °C, as shown in Figure 11e. 

### 4.3. Aqueous Electrolytes

SIBs based on aqueous electrolytes are safe, environmentally friendly, inexpensive, and less corrosive; thus, they are considered for different perspectives and applications [72]. Simultaneously, the aqueous electrolyte is also investigated and proved to have remarkable low-temperature performance. In 1966, Havemeyer [73] found that the freezing point of the water–DMSO mixture (cDMSO = 0.30) was considerably lower than the freezing points of both DMSO (18.9 °C) and water (0 °C) which was about −140 °C. Therefore, Zhanliang Tao et al. [74] synthesized and measured the freezing point of the DMSO system through differential scanning calorimetry (DSC), proving that the freezing point of the unique solvent is about −150 °C. Thus, it proved that adding DMSO (cDMSO = 0.30) in 2 M NaClO_4_ dramatically decreases the freezing point of the electrolyte and improving the low-temperature performance. At an extremely low temperature of −50 °C, the electrolyte conductivity reached 1.1*10^−4^ S/cm. Furthermore, the electrochemical function was tested by sandwiching the NaTi_2_(PO_4_)_3_@C (NTP@C) anode and carbon cathode. The full cell with the 2M-0.3 electrolyte exhibited an excellent specific capacity (ca. 68 mAh/g) and cycle performance (95%) over 100 cycles at a rate of 0.665 C. As demonstrated by Zhanliang Tao et al. [54], NaClO_4_ can be used as an electrolyte without any additives in extreme environments. Sandwiched by the NaTi_2_(PO_4_)_3_ anode and the nano Ni (OH)_2_ cathode, the full cell delivered favorable characteristics through a dual-ion reaction. It showed specific capacities of 82 mAh/g at 0 °C, 78 mAh/g at −10 °C, and 70 mAh/g at −20 °C, with an average voltage of 1.25 V. Moreover, the capacity retention remained 85% under a rate of 10 C after ultralong life of 10,000 cycles. Then, combined with SEM results, it can be concluded that there is no pulverization or agglomeration on the surface of the electrode at low temperatures. 

**Figure 11 nanomaterials-12-03529-f011:**
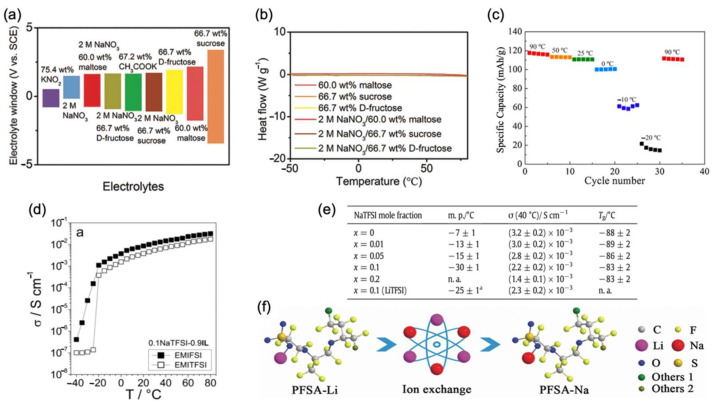
(**a**) Electrochemical windows for different electrolytes. (**b**) Heat flow of six sugar-based electrolytes in a wide temperature range. Reproduced with permission from Ref. [75]. Copyright 2020 WILEY-VCH Verlag GmbH & Co. KGaA, Weinheim, Germany. (**c**) Discharge capacities among a series temperature changes. Reproduced with permission from Ref. [69]. Copyright The Electrochemical Society of Japan. (**d**) EMIFSI and EMITFSI conductivity changes from −40 °C to 80 °C. Reproduced with permission from Ref. [68]. Copyright Creative Commons Attribution License. (**e**) PYR_14_TFSI-NaTFSI electrolytes’ melting point (m. p.), ionic conductivity (σ), glass change temperature (T_g_), and parameter (T_0_). (**f**) The acquirement of PFSA-N. Reproduced with permission from Ref. [71]. Copyright 2019 WILEY-VCH Verlag GmbH & Co. KGaA, Weinheim.

Madhavi Srinivasan et al. [66] introduced an ethanol-rich media electrolyte with a Na_0.44_MnO_2_ cathode and a Zn anode to reduce the irreversible proton co-insertion caused by aqueous media. The electrolyte consists of a 1 M sodium acetate in the 5:1 *v*/*v* ethanol–water solution compared with different ethanol–water volume ratios disclosed by contact angle tests (as shown in Figure 10d) and Raman scattering spectra (as shown in Figure 10e). They then drew the three-dimensional picture, as shown in Figure 10f, to further illustrate the ethanol–water system. It is a new type of hydrogen bond that effectively reduces the interfacial tension, resulting in the promoted contact between the interfaces of the two phases. Additionally, through the positive role in the rearrangement of the H_2_O network played by hydrogen bonding, the contact between the two phases is tighter and the signified dissolution of ions at low temperatures is decreased. Therefore, at 0 °C and 1 C rate, the full cell exhibited a specific capacity of 44.5 mAh/g with a potential of 1.2 V. After cycling for 50 cycles, the capacity retention still stayed at 94%. 

It is worth mentioning that a special hydroxyl-rich sugar solution was designed by Mianqi Xue et al. [75] to promote the original hydrogen bond in H_2_O. Thus, the sugar solution is able to reduce the proportion of free H_2_O molecules and break their original structure, resulting in less binding degree. After testing electrolytes containing different carbohydrates, it is found that super-concentrated sugars had wide electrolyte windows, reflecting its practicality in SIBs, as shown in Figure 11a. Moreover, this group tested sugar-based solutions in a wide temperature range from −50 °C to 80 °C, as shown in Figure 11b. It is proven that there is no obvious heat flow in both low or high temperatures, indicating that the sugar-based solution has the potential to be the electrolyte and their freezing points are all below −50 °C. In the group of Wu Zhongshuai et al. [76], a planar aqueous sodium-ion micro-battery was introduced in the water-in-salt electrolyte. The water-in-salt electrolyte matched perfectly with the NVP-based anode and cathode, which delivered a favorable voltage window of 2.7 V versus Na^+^/Na at an extremely low temperature of −50 °C. The system has already exhibited satisfactory stability in long-term cycling. After 1000 cycles at room temperature, it exhibited a capacity retention of 88%. Meanwhile, at −40 °C, its coulombic efficiency is still above 99% because of the interdigital in-plane geometry system.

### 4.4. Solid-State Electrolytes

Liquid electrolytes have safety hazards and flammability risks when vehicles are hit or the temperature rises sharply which has been leaked in lithium-ion battery energy vehicles. The lithium-ion battery car combustion incidents that have occurred in recent years have brought people’s concerns back to battery safety. Furthermore, the higher viscosity of the electrolyte at low temperatures causes more severe polarization, which is an important reason for the electrochemical performance degradation of liquid electrolytes. Thus, solid-state sodium batteries (SSIBs) have emerged as an attractive choice to solve these problems. Maowen Xu et al. [77] prepared a polymer-based solid-state sodium electrolyte (PFSA-Na membranes) by dissolving the PFSA-Na powder in N, N-dimethyl-formamide. As shown in Figure 11f, PFSA-Na was obtained by a facile ionic exchange method, replacing Li^+^ at the end of the chain by Na^+^. Then, after stirring to a homogeneous solution, EC/DEC with 1 M NaClO_4_ was added. After stirring, the PFSA-Na membranes were finally obtained by a facile solution blade coating method. PFSA-Na membranes demonstrated high ionic conductivity of 4.88*10^−5^ S/cm even at −15 °C. Assembled with Prussian blue, the half-cell showed superior specific capacities of 100 mAh/g at −5 °C, 93 mAh/g at −15 °C, and 80 mAh/g at −20 °C at a rate of 1 C. They also tested the low-temperature characteristics of the PFSA-Na electrolyte and reported the ideal electrochemical performance of the PFSA-Na membrane (PFSA-Na powder dissolved in DMF and added with 1 M NaClO_4_ in EC/DEC). Figure 12a shows that PFSA-Na has a conductivity of 4.82*10^−5^ S/cm, which is very similar to that of Xu’s work. Assembled with the polyanionic cathode Na_3_V_2_O_2_(PO_4_)_2_F and the PFSA electrolyte, the cell shows a highly reversible sodium-ion extraction–insertion process, resulting in remarkable low-temperature electrochemical performance. Meanwhile, due to the perfect match among the NASICON cathode and the PFSA-Na membrane, the system shows limited polarization. The half-cell delivered suitable discharge capacities, high voltage, and stable low-temperature performance, as mentioned in the cathode part. 

In another group, Guanglei Cui et al. [78] reported a new process to synthesize the quasi-solid electrolyte, with poly(methyl vinyl ether-alt-maleic anhydride) (P(MVE-alt-MA)) as the host, including three steps. First, they dissolved P(MVE-alt-MA) into acetonitrile to form the host of the electrolyte. Then, they poured the solution into a dry bacterial cellulose (BC) for reinforcement. When it dried out, they cut it into circles and soaked it in the 1 M NaClO_4_/TEP-VC electrolyte which acted as a plasticizer. Finally, the target electrolyte was obtained. It was proven that Na dendrites, which seriously impact SEI formation, are hindered by quasi-solid P(MVE-alt-MA) electrolytes. Moreover, the investigation elucidated that it could promote the synthesis of an interface between the NVP cathode and the electrolyte, which suppressed the vanadium’s dissolution and enhanced the batteries’ safety. Therefore, when assembled with the NVP anode, the half-cell exhibited stable electrochemical performance in low temperatures. A 70 mAh/g specific capacity was achieved at −10 °C with a retention capacity of 84.8% after 50 cycles at 0.1 C, as shown in Figure 12b. In addition, Zhong-Shuai Wu et al. [79] applied photopolymerization to develop an ethoxylated trimethylolpropane triacrylate-based quasi-solid-state electrolyte (QSSE). The QSSE is obtained by mixing 1 M NaClO_4_ in the PC and 5% FEC into the photocuring agent ethoxylated trimethylolpropane triacrylate (ETPTA) and 1% 2-hydroxy-2-methylpropiophenone, before being dropped on a cellulose film and cured with ultraviolet light. Due to the resultant, the QSSE exhibited remarkable ionic conductivity of 7*10^−4^ S/cm at a low temperature of 0 °C and 8*10^−4^ S/cm at a low temperature of −10 °C.

**Figure 12 nanomaterials-12-03529-f012:**
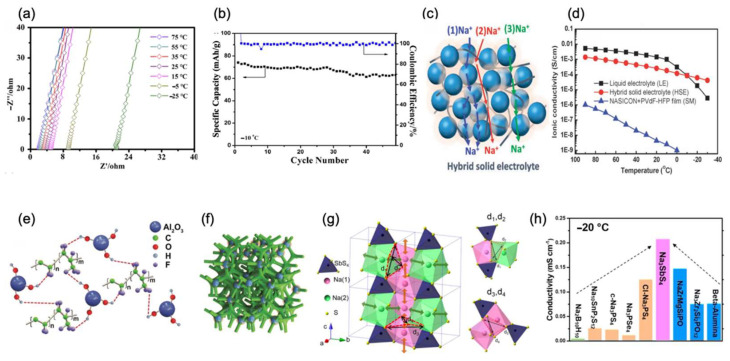
(**a**) AC impedance spectra of PFSA-Na at different temperatures. Reproduced with permission from Ref. [77]. Copyright 2020 Wiley-VCH GmbH. (**b**) Electrochemical performance of NVP/Na half-cell using (flame-retardant quasi-solid polymer electrolyte) FRPMM-CPE after 50 cycles at −10 °C. Reproduced with permission from Ref. [78]. Copyright 2018 American Chemical Society. (**c**) Na^+^ hopping pathways in the hybrid solid electrolyte. Reproduced with permission from Ref. [80]. Copyright The Royal Society of Chemistry 2015. (**d**) Ionic conductivity comparison of the solid film, liquid electrolyte, and hybrid solid electrolyte. (**e**) Constitutional unit of PVDF-HFP and Al_2_O_3_ nanoparticles. (**f**) Three-dimensional schematic illustration of the PVDF-HFP three-dimensionally cross-linked structure. Reproduced with permission from Ref. [81]. Copyright 2019 WILEY-VCH Verlag GmbH & Co. KGaA, Weinheim. (**g**) Three-dimensional model of the Na_3_SbS_4_ framework. Structure of the tetragonal Na3SbS4 framework (note: the relative radii of spheres do not represent the real atom sizes) with conduction paths of Na(1)-Na(1) (along z-axis) and Na(1)-Na(2) (in xy-plane). The right schematic display is magnified to show the S-gates (d_1_, d_2_, d_3_, d_4_) and the contractions. (**h**) Comparison of the solid Na_3_SbS_4_ electrolyte with the most reported solid electrolytes. Reproduced with permission from Ref. [82]. Copyright 2018 American Chemical Society.

In another work of Youngsik Kim et al. [80], they introduced Na_3_Zr_2_Si_2_PO_12_-based composite to form polymer electrolyte. As for the constitution, the weight percentage ratio is 70: 15: 15. Here, 70% is the Na_3_Zr_2_Si_2_PO_12_-based composite, 15% is poly (vinylidene fluoride-co-hexafluoropropylene) (abbreviate as PVdF-HFP), and 15% is 1 M NaCF_3_SO_3_/TEGDME. Meanwhile, when the temperature goes down, PVdF-HFP effectively promotes the stability of the interface of two solid materials. The application of ceramic powder not only plays a similar role as the PVdF-HFP binder, but it also facilitates the transformation of Na^+^, resulting in high conductivity in low temperatures. Unlike at low current densities, the addition of strong ionic forces to the conductive ceramic at high current densities can facilitate the transfer of sodium ions in the electrolyte shell, resulting in the promising electrochemical performance of conductive ceramics at high rates, as shown in Figure 12c. Thus, the electrolyte is demonstrated to possess superior low-temperature ionic conductivity of 3*10^−4^ S/cm at 0 °C, 1*10^−4^ at −10 °C, and 2*10^−5^ S/cm at 0 °C, as shown in Figure 12d. Furthermore, Yongbing Tang et al. [81] developed a flexible sodium-based dual-ion battery based on a QSSE, consisting of PVDF-HFP three-dimensionally cross-linked with Al_2_O_3_ nanoparticles (Figure 12e,f), which exhibits a porous 3D structure with dramatically enhanced low-temperature electrochemistry. Compared with liquid electrolyte, it is reported that QSSE has superior electrochemical performance and more favorable stability during a series of temperature changes in SIBs. Furthermore, the QSSE facilitates the fast ionic migration of both anions and cations. Thus, sandwiched by the graphite cathode and the Sn anode, the full cell showed specific capacities of 65 mAh/g at −5 °C and 45 mAh/g at −20 °C with rate of 5 C. Moreover, Chengdu Liang et al. [82] devised a 3D superionic conductor Na_3_SbS_4_ solid electrolyte model to investigate a conductive mechanism, as shown in Figure 12g. Because of the great structural stability with the 3D conductive tunnel network, the ionic conductivity would only slightly drop in an extreme low-temperature environment. Therefore, at −20 °C, Na_3_SbS_4_ was reported to hold an ionic conductivity of 2*10^−4^ S/cm, which is the highest among the most reported solid electrolytes, as shown in Figure 12h.

In the end of this section, we list the electrolytes mentioned above in Table 1. Table 1 compares the conductivities of all the above-mentioned electrolytes at low temperatures, the specific capacities, and the number of battery cycles. Electrolytes always easily undergo phase change and more complex interface evolution at low temperatures, leading to a decrease in the cycle stability of the battery. Therefore, the number of working cycles of the battery was taken as an important index. As the research proceeds, a combination of an anode, a cathode, and an electrolyte of sodium-ion battery is being constantly improved. Even at low temperatures, most of the electrolyte systems have an ionic conductivity of 10^−3^~10^−4^ S/cm, comparable to that of the electrolyte or commercial diaphragm system at normal temperatures. These electrolytes into the batteries show ideal electrical properties and stability. Moreover, some of SIBs have the capacity retention of more than 80% after 500 cycles, showing the application potential of sodium-ion batteries at low temperatures. The NaClO_4_- and NaPF_6_-based organic electrolytes are widely used in practical applications. Compared to the conventional organic electrolyte, the current research on aqueous electrolytes, ionic liquid electrolytes, and quasi-solid and solid electrolytes at low temperatures is just focused on ionic conductivity. Therefore, later research should focus more on the practical application of electrolytes, such as the adaptability of electrolytes with a cathode and anode at low temperatures. Meanwhile, the evolution of interfaces between different polymer skeletons, solvent molecules, and electrodes is so complex at low temperatures, making it necessary to structurally characterize and systematically understand SEI films.

## 5. Conclusions

To sum up, sodium, as a substitute for lithium, is used as an energy storage material and is abundant in the earth’s surface. In SIBs, Na^+^ has higher ionic conductivity than Li^+^. SIBs can use aluminum as a cathode, and are lighter and safer when used on a large scale. However, in the research of sodium-ion batteries, its low-temperature performance is still a problem. This paper discusses the improved low-temperature performances of sodium-ion batteries from three aspects: cathode material, anode material, and electrolyte.

PB and PBAs are widely used as traditional cathode materials in SIBs. Combined with CNTs, the ionic conductivity of cathode material was enhanced, and the charge–discharge specific capacity and cycle stability were also promoted. Simultaneously, some special cathode structures, such as cubic nanoparticles and crystalline particles, also make the batteries, showing ideal stability at low temperatures. Then, layered oxides composed by transition metals exhibit favorable electrochemical performance. Further structural modifications, such as NaTi_2_(PO_4_)_3_ coating, NaCrO_2_ nanowire fabrication, or hierarchical columnar structure design, can reduce the attenuation of battery performance at low temperatures. As for polyanionic-type cathodes, C coating, CNTs, and C nano wares are also effective ways of advancing low-temperature cathode performance. By comprehensively considering cathode performance at low temperatures, many materials have a specific capacity of about 100 mAh/g and a relatively stable voltage window of about 3 V, showing bright application potential.

HC is mostly used as an anode material in SIBs. At low temperatures, some effective modifications to HC can also play a role in promoting its performance. Similar to cathodes, in some special ways, such as C coating or structure-like micropores, HC can increase the conductivity of the anode to obtain an ideal battery-specific capacity. Amorphous selenium and metal selenides, on the other hand, use Se in combination with the metal elements Zn, Fe, or Sn as the anode, and can obtain a lower charge and discharge potential after modification with CNTs or graphite. NaTi_2_(PO_4_)_3_ can also obtain favorable electrochemical properties by combining sp2-type carbon or high-crystallinity NaV_1.25_Ti_0.75_O_4_ in the anode. Considering representative materials, we found that HC is still the material with the lowest voltage and largest specific capacity.

In the end, under low-temperature conditions, the Na^+^ transfer rate in the electrolyte is slowed down, and the ion conductivity of the electrolyte and the diffusion rate of Na^+^ in the active material will be reduced. The charge transfer process is difficult at the solid–liquid phase interface on the surface of material. In the four electrolytes mentioned in this paper, the traditional organic electrolytes need to be mixed with organic mixtures of different proportions to improve its efficiency. The ionic liquid electrolytes have the highest ion transfer rate and dielectric constant. Aqueous electrolytes also have ideal conductivity at low temperatures and are environmentally friendly. Solid-state electrolytes have relatively safe qualities, can effectively avoid the fire hazard on organic electrolytes, and have been proven to have practical low-temperature performance. This article provides feasible suggestions for the future development of SIBs at low temperatures.

## Figures and Tables

**Figure 1 nanomaterials-12-03529-f001:**
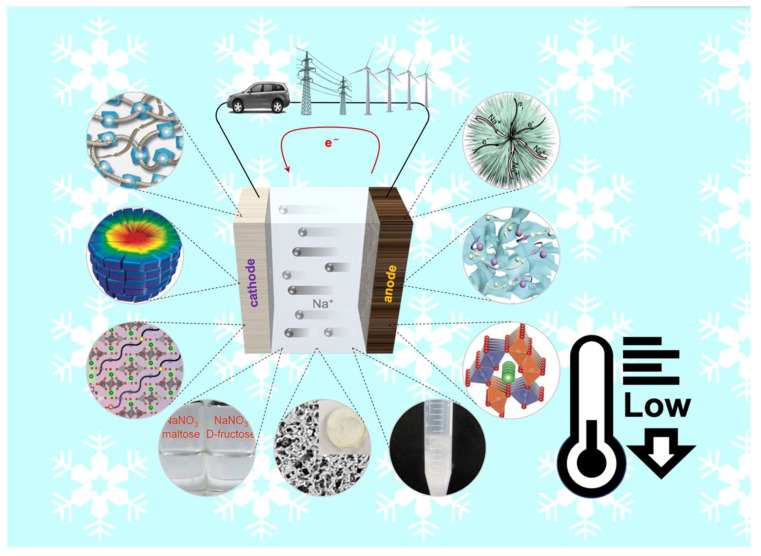
Schematic diagram of sodium-ion battery at low temperatures.

**Figure 2 nanomaterials-12-03529-f002:**
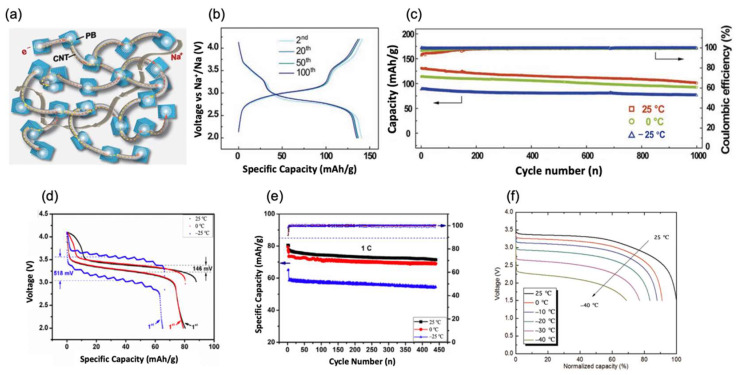
(**a**) The system of PB/CNT with carbon nanotube connecting PB. (**b**) Cycling performance of PB/CNT at a current density of 0.1 C for a long run at −25 °C. (**c**) PB/CNT has an ultra-long cycle life at a high rate of 2.4 C at 25 °C, 0 °C, and −25 °C. Reproduced with permission from Ref. [18]. Copyright 2016 WILEY-VCH. (**d**) PBNi-ES temperature comparison for voltage and specific capacities at 0.1 C. (**e**) High rate of 1 C and long cycle performance of PBNi-ES. Reproduced with permission from Ref. [19]. Copyright 2018 Elsevier. (**f**) Cell performance at a range of temperatures from an ambient temperature to an extremely low temperature of −40 °C at a 1 C rate. Reproduced with permission from Ref. [21]. Copyright 2018 WILEY-VCH.

**Figure 4 nanomaterials-12-03529-f004:**
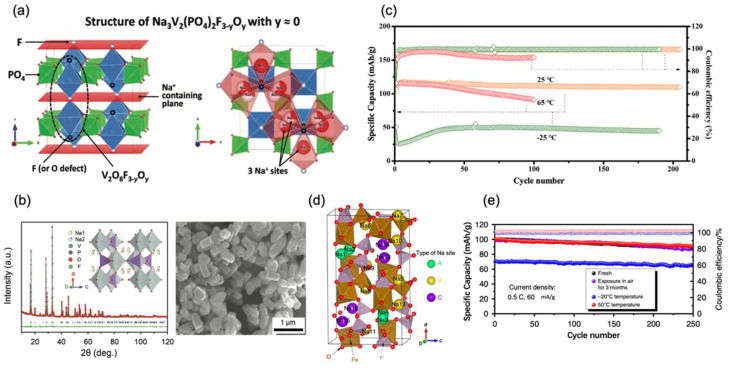
(**a**) On the left, the 3D schematic illustration of NVPF_3−y_O_y_. The black dots represent the transformation of oxygen and fluorine. The red sheet indicates Na^+^ filling. On the right site, Na^+^ fills into the voids of oxygen and fluorine. Reproduced with permission from Ref. [38]. Copyright 2018 WILEY-VCH. (**b**) XRD image, schematic illustration, and SEM image of NVPF-NTP material. Reproduced with permission from Ref. [39]. Copyright 2017 WILEY-VCH. (**c**) Long-term cycling performance of Na//PFSA-Na//NVOPF@rGO half-cells at a high rate of 1 C and extremely low temperatures. Reproduced with permission from Ref. [40]. Copyright 2020 Wiley-VCH. (**d**) Schematic illustration of crystal structural Na_4_Fe_3_(PO_4_)_2_(P_2_O_7_) material. (**e**) Long life span of 250 cycles for Na_4_Fe_3_(PO_4_)_2_(P_2_O_7_)/C at −20 °C. Reproduced with permission from Ref. [41]. Copyright 2019, Springer Nature Limited.

**Figure 5 nanomaterials-12-03529-f005:**
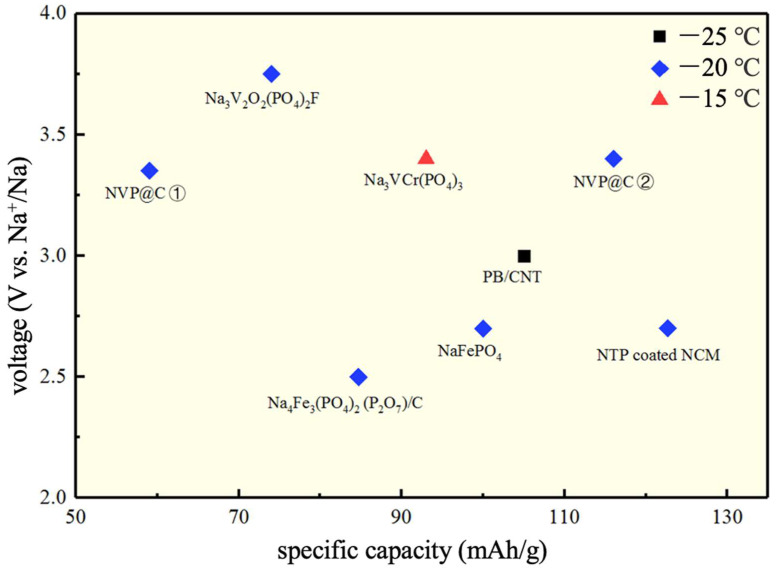
Eight different representative cathode materials in low temperatures which are carbon-coated Na_3_V_2_(PO_4_)_3_ and conventional organic electrolytes at a 1 C rate (NVP@C① in Figure 5) [33], hybridized Na_3_V_2_O_2_(PO_4_)_2_F tested at a 1 C rate [40], Na_3_VCr(PO_4_)_3_ tested at 0.1 C [29], carbon-coated NaFePO_4_ at 0.1 C [28], Na_2_Fe[Fe(CN)_6_] and carbon nanotubes at a 0.1 C rate [18], carbon-coated Na_3_V_2_(PO_4_)_3_ and ionic liquid electrolytes (NVP@C② in Figure 5) at a 1 C rate [31], and NaTi_2_(PO_4_)_3_-coated Na_0.67_Co_0.2_Mn_0.8_O_2_ at a 0.2 C rate [22], from left to right, respectively.

**Figure 6 nanomaterials-12-03529-f006:**
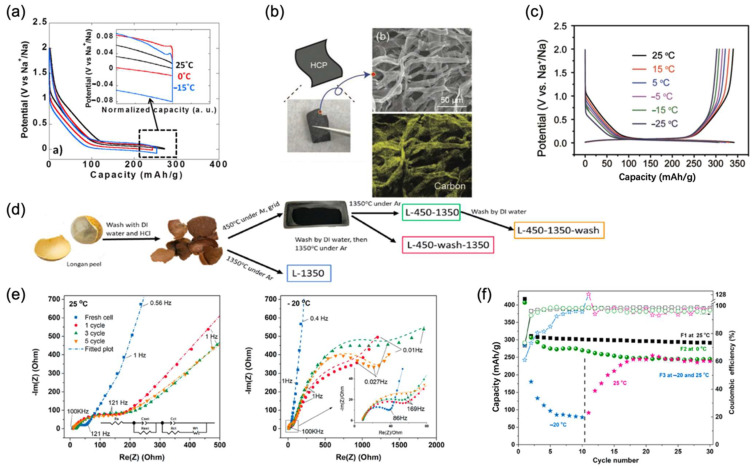
(**a**) Electrochemical performance of carbon-covered HC. The black line represents the anode tested at 25 °C, the red line represents the anode tested at 0 °C, and the blue line represents the anode tested at −15 °C. Reproduced with permission from Ref. [42]. Copyright 2015 Elsevier. (**b**) SEM image and direct photo of HCP and its inner structure. (**c**) Electrochemical performance of HC nano paper at 0.5 C from −25 to 25 °C. Reproduced with permission from Ref. [43]. Copyright 2019 WILEY-VCH. (**d**) Schematic of the longan peel process to produce HC. (**e**) HC anode Nyquist plot image at 25 °C and −20 °C. (**f**) Comparison of electrochemical performance when the temperature changes. Nyquist plot of half-cells at 25 °C and −20 °C. The dash lines represent the fitted plot from the equivalent circuit. Reproduced with permission from Ref. [44]. Copyright 2019 Elsevier.

**Figure 9 nanomaterials-12-03529-f009:**
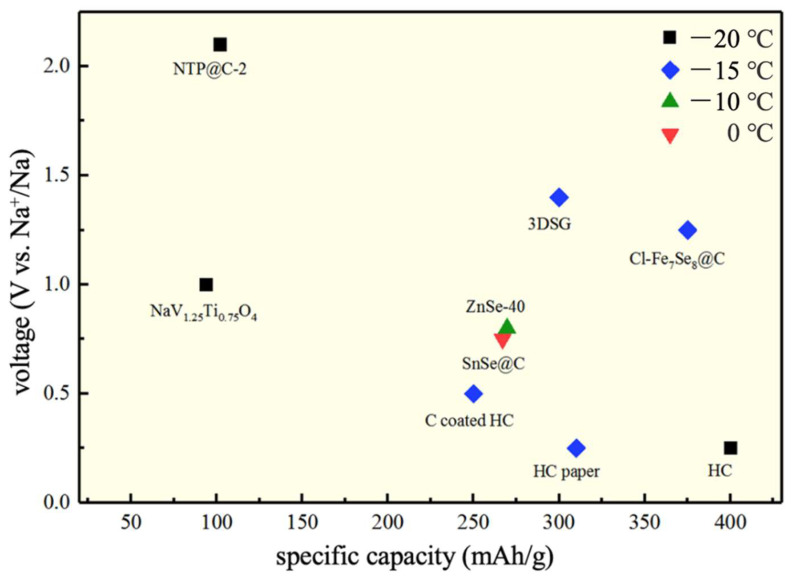
Nine different representative anode materials in low temperatures which are NaV_1.25_Ti_0.75_O_4_ at a 0.2 C rate [55], carbon-covered NaTi_2_(PO_4_)_3_ with Na_3_V_2_(PO_4_)_3_ at a 0.2 C rate [52], carbon-coated HC at a 0.1 C rate [42], SnSe@carbon nanofibers at a 1 C rate [48], ZnSe@carbon nanotubes at a 1 C rate [49], amorphous selenium-coated reduced graphene oxide nanosheets at a 2 C rate [46], HC paper and carbon microbelts at a 0.5 C rate [43], coral-like Fe_7_Se_8_@carbon nanorods at a 0.5 C rate [50], and HC at a 0.1 C rate [44], from left to right, respectively.

**Table 1 nanomaterials-12-03529-t001:** Different electrolytes and their main low-temperature electrochemical performances.

Electrolytes	Temperature (°C)	Conductivity (S/cm)	Specific Capacity (mAh/g)	Retention Capacity/Number of Cycles	Anode//Cathode
1 M NaClO_4_ in PC 2 vol% FEC Ref. [55]	−20	/	94 0.2 C		Na//NaV_1.25_Ti_0.75_O_4_
	−20		93 0.2 C	84%/200 1 C	Na_0.8_Ni_0.4_Ti_0.6_O_2_//NaV_1.25_Ti_0.75_O_4_ full cell
1 M NaClO_4_ PC with 5 vol % FEC Ref. [28]	−10	/	113 0.2 C	93%/200 0.5 C	Na//NFP@C
	−20		100 0.2 C	94%/200 0.5 C	
1 M NaClO_4_ in EC/PC 5wt% FEC Ref. [46]	−5	/	375 20 C	96.2%/1000 20 C	Na//3DSG composite
	−15		300 20 C	98%/1000 20 C	
	−25		250 20 C	90.4%/1000 20 C	
1 M NaClO_4_ in EC/PC 5wt% FEC Ref. [50]	0	/	425 2 C	96%/80 2 C	Na//cl-Fe_7_Se_8_@C
	−15		375 2 C	82%/80 2 C	
	−25		350 2 C	65%/80 2 C	
1 M NaClO_4_ in EC/DMC 1:1 (vol) Ref. [19]	0	/	79.3 0.5 C	87%/440 1 C	Na//PBNi-ES
	−25		65.1 0.5 C	84%/440 1 C	
1 M NaPF_6_ EC/PC 1:1 (wt) 10wt% DMC Ref. [42]	0	/	245 0.1 C	/	HC@C//Na
	−15		150 0.1 C		
1 M NaClO_4_ EC/DEC 1:1 (vol) Ref. [52]	0	/	105 0.266 C	/	Na//NTP@C-2
	−10	/	110 0.266 C		
	−20	/	102 0.266 C		
1 M NaClO_4_ EC/PC 1:1 (vol) Ref. [6]	0	/	155 0.3 C	81%/1000 2.4 C	Na//PB/CNT
	−25		105 0.3 C	86%/1000 2.4 C	
0.5 M NaPF_6_ EMS: FEC 98:2 (vol) Ref. [18]	0	/	130 0.75 C	89%/100 0.75 C	HC//RAHC
	−20		100 0.75 C	92%/100 0.75 C	
1 M NaClO_4_ EC/PC 1:1 (vol) Ref. [31]	0	/	120 0.2 C		Na//NVP@C
	−10		120 0.2 C	97.2%/40 1 C	
	−20		116 0.2 C	75.8%/500 10 C	
molar fraction 0.3% DMSO 2 M NaClO_4_ Ref. [74]	−50	1.1 × 10^−4^	68 0.665 C	95%/100 0.665 C	NaTi_2_(PO_4_)_3_@C//activated carbon
2 M NaClO_4_ Ref. [54]	0	/	82 10 C	85%/10,000 10 C	NaTi_2_(PO_4_)_3_@C//nanoNi(OH)_2_
	−10		78 10 C		
	−20		70 10 C		
1 M sodium acetate in ethanol/water 5:1 (vol) Ref. [66]	0	/	44.5	94%/50 1 C	Zn//Na_0.44_MnO_2_
0.2NaFSI-0.8EMIFSI Ref. [33]	−10	/	78.1 0.1 C	/	Na//NASICON-NVP@C
	−20		58.6 0.1 C		
0.2NaFSI-0.8C_3_C_1_pyrrFSI Refs. [69,70]	0	9.8 × 10^−4^	100 0.2 C	95%/500 1 C	Na//NaCrO_2_
	−10	/	60 0.2 C	/	
	0	/	80 0.2 C	/	Na//Na_2_FeP_2_O_7_
	−10		67 0.2 C		
	−20		42 0.2 C		
0.1NaTFSI-0.9EMIFSI Ref. [68]	−20	1.1 × 10^−3^	/	/	/
0.1NaTFSI-0.9EMITFSI Ref. [68]	−20	3.8 × 10^−4^	/	/	/
0.1NaTFSI-0.9PYR_14_TFSI Ref. [71]	−30	2.2 × 10^−3^	/	/	/
25m NaFSI-10m NaFTFSI Ref. [53]	−10	/	65 0.2 C	89%/500 0.2 C	NaTi_2_(PO_4_)_3_//Na_3_(VOPO_4_)_2_F
PFSA-Na membrane (PFSA-Na/DMF solution with 1 M NaClO_4_ in EC/DEC) Ref. [77]	−5	/	100	/	Na//HQ-NaFe
	−15	4.88 × 10^−5^	93	/	
	−25	/	80	/	
host: P(MVE-alt-MA) reinforce: BåC plasticier: TEP/VC/NaClO_4_ abbreviated FRPMM-CPE Ref. [78]	−10	/	70 0.1 C	84.8%/50 0.1 C	Na//Na_3_V_2_(PO_4_)_3_
	−15		61.6 0.05 C	/	
PFSA-Na membrane (PFSA-Na powder dissolved in DMF and added with 1 M NaClO_4_ in EC/DEC) Ref. [40]	−5	/	107 1 C	/	Na//PFSA-Na//Na_3_V_2_O_2_ (PO_4_)_2_F
	−15		84 1 C		
	−20		74 1 C		
	−25	4.82 × 10^−5^	58 1 C	99.6/190 1 C	
	−25		40.6 1 C	90%/30 1 C	HC//PFSA-Na//Na_3_V_2_O_2_ (PO_4_)_2_F
PVdF-HFP binder, Na_3_Zr_2_Si_2_PO_12_-based composite, and 1 M sodium triflate NaCF_3_SO_3_/TEGDME liquid electrolyte in 7:1.5:1.5 (wt) Ref. [80]	0	3 × 10^−4^	/	/	/
	−10	1 × 10^−4^			
	−20	2 × 10^−5^			
Na_3_SbS_4_ solid electrolyte Ref. [82]	−20	2.2 × 10^−4^	/	/	/
PVDF-HFP membrane with 5 wt% of Al_2_O_3_ nanoparticles Ref. [82]	−5	/	65 5 C	/	Sn//graphite
	−20		45 5 C		
ETPTA-NaClO_4_-QSSE Ref. [79]	0	7 × 10^−4^	/	/	/
	−10	8 × 10^−4^			

## Data Availability

Not applicable.

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
