# Peer review of "Recent Progress and Perspective: Na Ion Batteries Used at Low Temperatures"

_nanomaterials, 2022, doi:10.3390/nano12193529_

Round 1

Reviewer 1 Report

This review is focusing on the materials development for sodium-ion batteries with low-temperature applications. A development and recent research for positive / negative electrode materials and electrolytes are summarized by three parts of sections. Some details are also described in subsections in each. Overall, the manuscript is well configurated and the structure is readable. This review will be interesting to the readers of nanomaterials as well as the researchers of energy storage devices applications. However, some minor revisions as follows should be better to addressed before publication.

[1] Some published data of electrode materials are scattered in the main text, are decreasing of readabilities. They should be summarized as tables on applicable sections.

[2] In the introduction section

“In terms of large-scale energy storage, SIBs are safer than LIBs [13].”

Generally, safety issues of batteries are depending on the various cell configurations and operating conditions. Therefore, it could not say that SIBs are safer than LIBs in any condition. The description should be revised better, such as SIBs are safe as well as LIBs.

[3] In the introduction section

“SIBs can discharge completely, so that there is no need to worry about over-discharge.”

Over-discharge is a matters of battery operation and not to directly connect the single-cell performance. The sentence should be revised such as, stable discharge performance of SIBs cell makes it easy to manage of the depth of discharge.

[4] Some abbreviations are used without explanations of formal expressions. Such as DI, HC, ANIMBs, WiS and QSSE. They should be described with formal name at firstly appeared.

[5] Figure 5 and Figure 9

The labels of vertical axis should be revised as Voltage (V vs. Na+/Na).

[6] Section 3.1

The safety issues of hard-carbon anode materials with low-temperature applications, such as dendrite formation, will be better to added.

[7] Section 3.2

The reason that focused on the Se-based materials should be expressed more detail. Are there any specific advantageous on that materials than any other alloy-based materials such as P or Sn.

[8] Section 3.3

Some disadvantageous of NaTi2(PO4)3 material, such as low energy density as anode materials, will be better to explain.

[9] Section 3.3

Other titanium-based anode materials, such as TiO2 and Na2Ti3O7 should be introducing on the text or figure.

[10] Conclusion

First paragraph is mostly duplicate to some part of abstract. In the conclusions, the main focus of this review should be summarized.

Author Response

Response to Reviewer #1

General comment: This review is focusing on the materials development for sodium-ion batteries with low-temperature applications. A development and recent research for positive / negative electrode materials and electrolytes are summarized by three parts of sections. Some details are also described in subsections in each. Overall, the manuscript is well configurated and the structure is readable. This review will be interesting to the readers of nanomaterials as well as the researchers of energy storage devices applications. However, some minor revisions as follows should be better to addressed before publication.   

Response: We thank the reviewer for the effort in improving the quality of our manuscript.

  1. Some published data of electrode materials are scattered in the main text, are decreasing of readabilities. They should be summarized as tables on applicable sections.

Response: Thanks for your suggestion. In the electrolyte section, we summarized published data of electrode materials in the table (Table 1).  

  1. In the introduction section

“In terms of large-scale energy storage, SIBs are safer than LIBs [13].”

Generally, safety issues of batteries are depending on the various cell configurations and operating conditions. Therefore, it could not say that SIBs are safer than LIBs in any condition. The description should be revised better, such as SIBs are safe as well as LIBs.

Response: We are sorry for this confusion. We have changed the description in the revision. (Page 2 Line 12-14)

  1. In the introduction section

“SIBs can discharge completely, so that there is no need to worry about over-discharge.”

Over-discharge is a matters of battery operation and not to directly connect the single-cell performance. The sentence should be revised such as, stable discharge performance of SIBs cell makes it easy to manage of the depth of discharge.

Response: We are sorry for this confusion. We have changed the description in the revision. (Page 2 Line 12-14)

  1. Some abbreviations are used without explanations of formal expressions. Such as DI, HC, ANIMBs, WiS and QSSE. They should be described with formal name at firstly appeared.

Respond:Yes. We have described them when they firstly appeared in the revision. 

  1. Figure 5 and Figure 9

The labels of vertical axis should be revised as Voltage (V vs. Na+/Na).

Response: Yes. Voltage (V vs. Na+/Na) has already been added in Figure 5 and Figure 9 in the revision.

  1. Section 3.1

The safety issues of hard-carbon anode materials with low-temperature applications, such as dendrite formation, will be better to added.

Response: We added safety issues of hard-carbon anode materials with low-temperature applications in the revision. (Page 11, Line 11-13)

  1. Section 3.2

The reason that focused on the Se-based materials should be expressed more detail. Are there any specific advantageous on that materials than any other alloy-based materials such as P or Sn.

Response: We are sorry for this confusion. We have briefly added advantages of Se-based materials in the revision (Part 3.2 paragraph 1).

  1. Section 3.3

Some disadvantageous of NaTi2(PO4)3 material, such as low energy density as anode materials, will be better to explain.

Response: We are sorry for this confusion. We have briefly disadvantages of NaTi2(PO4)3 in the revision (Page 15 Line 1-15).

  1. Section 3.3

Other titanium-based anode materials, such as TiO2 and Na2Ti3O7 should be introducing on the text or figure.

Response: Thanks for your suggestion. Research on titanium-based anode at low temperatures are extremely few, common materials such as TiO2 and Na2Ti3O7 are not reported as anode material at low temperatures yet, so we are unable to provide more information about titanium-based anode materials in this review.

  1. Conclusion

First paragraph is mostly duplicate to some part of abstract. In the conclusions, the main focus of this review should be summarized.

Response: Yes. We have rewritten the abstract part. The purpose of this review is to provide some basic information of cathode, anode and electrolyte of SIBs at low temperatures, we have summarized the main focus of this review in the revision.

Reviewer 2 Report

The paper in question has been positioned as a review. The choice of the topic cannot be criticized, since the subject itself (metal-ion batteries) is extremely important, the sub-subject (sodium-ion batteries) is probably the most important branch at the current state of the art, and the particular issue (low-temperature behavior) is among the hot ones. In this regard, the choice of the review topic is excellent.

The review is well-organized at the top level. I particularly liked the chapters on cathode, anode, and electrolyte discussed separately (however, I have got an argument about this). Unfortunately, within each of the chapters I found little system in the discussion of the published results and the related approaches. Are the first/last examples most common? most striking? most recent? So far, each of the chapters looks an collection of relevant papers listed in an arbitrary order. Moreover, I have found very little opinion from the authors of the review (for me, review means more than a compilation of others' reports, it should bring some idea, judgement about promises of this or that approach, comparison of the approaches, general physical and chemical ideas behind them, etc.). The presented examples are very 'heterogeneous' in the approaches reported in the original papers - starting from the chemistry and ending at the nanostructure of the material, the scope of the journal the review has been submitted to. To make the things even worse, citation of the results from the original papers is often incomplete and\or nonuniform. Here and there, (dis)charge rate is omitted when discussing the capacity or capacity retention at different temperatures. The (dis)charge rate in the 'C' scale and in the A/g units are discussed simultaneously sometimes, which makes it difficult to compare the figures.

Finally, the conclusion is too broad (which is not surprising, since there has been little thought from the review authors throughout the text). Which approach(es) listed in the review the authors consider the most promising? most questionable and needing verification? and why? An important issue which has been completely missed is that although it is useful to separate the battery into parts to carry out a focused research, the whole battery (which is an ultimate goal in this field) is not just a sum of the components. Are there any signs that this or that type of electrolyte, being superior as such, performs worse in combination with particular type of anode/cathode? Or, vice versa, maybe some combination of otherwise moderate cathode, anode, and electrolyte can suddenly provide an excellent battery when assembled? Or the components can indeed be improved separately? For me, this question (which can be answered upon accurate comparative analysis of the data cited by the authors) is even more important than actual capacities, numbers of cycles, charge rate, etc achieved in the original papers.

To sum up, unfortunately at this point I cannot consider this a nice review, but rather a compiled collection of original papers divided into three groups. Therefore, my recommendations to the authors are as follows:

1. Decrease the number of citations or extend your own discussion (especially in the introduction - there are 28 references per 37 lines of text - which means that there are hardly 10-15 words explaining each reference!)

2. Consider adding more appropriate discussion of each of the examples cited - or at least the most important ones. Include more generalization and provide your own opinion regarding the most promising/facile/cheap approaches to be considered in future research.

3. Please be more accurate in the citation of the original results. Provide all the information needed to properly judge about the importance of the study - the most important piece is the information on (dis)charged rate missing in many of the citations.

4. I would suggest adding the information about the rate in Figs. 5 and 9 (probably coded by the symbol size).

5. Remove Fig. 10 - it adds nothing important to the listing in the text. It could be relevant in an oral presentation, but it adds nothing to a paper.

6. Please accurately check every letter in the paper. Besides grammar errors, there are small yet inaccurate things: for example, incorrect units in line 49, 'Figured' in line 67, subscripts in lines 120 and 123, units in line 195, 'anode' instead of 'cathode' in line 287, 'former' in line 401, the name in line 440, 'imidazolyl' instead of 'imidazolium' in line 624, obsolete hyphens in the chemical names in lines 640 and 654, etc. The list is not complete, there are just examples of numerous inaccuracies. 

7. There are too many abbreviations, some of which are quite confusing (for example, NMP in the field of batteries conventionally stands for N-methylpyrrolidone rather than to a special type of electrode material). Moreover, some of the abbreviations are introduced well after their first use (for example, HC for hard carbon is introduced in line 308, being used in line 142). Some of the abbreviations are never explained (for example, ES in line 89). Therefore, I recommend the authors to add a separate section with the abbreviations used in the review.

8. Regarding lines 577-579. I do not see how the optical images allow the conclusion on the viscosity. Please explain in more detail.

I hope that addressing these points will make the review more readable (and it will in fact become a review rather than a collection of references). Good luck! 

Author Response

Response to Reviewer #2

General comment: The paper in question has been positioned as a review. The choice of the topic cannot be criticized, since the subject itself (metal-ion batteries) is extremely important, the sub-subject (sodium-ion batteries) is probably the most important branch at the current state of the art, and the particular issue (low-temperature behavior) is among the hot ones. In this regard, the choice of the review topic is excellent.

The review is well-organized at the top level. I particularly liked the chapters on cathode, anode, and electrolyte discussed separately (however, I have got an argument about this). Unfortunately, within each of the chapters I found little system in the discussion of the published results and the related approaches. Are the first/last examples most common? most striking? most recent? So far, each of the chapters looks a collection of relevant papers listed in an arbitrary order. Moreover, I have found very little opinion from the authors of the review (for me, review means more than a compilation of others' reports, it should bring some idea, judgement about promises of this or that approach, comparison of the approaches, general physical and chemical ideas behind them, etc.). The presented examples are very 'heterogeneous' in the approaches reported in the original papers - starting from the chemistry and ending at the nanostructure of the material, the scope of the journal the review has been submitted to. To make the things even worse, citation of the results from the original papers is often incomplete and\or nonuniform. Here and there, (dis)charge rate is omitted when discussing the capacity or capacity retention at different temperatures. The (dis)charge rate in the 'C' scale and in the A/g units are discussed simultaneously sometimes, which makes it difficult to compare the figures.

Finally, the conclusion is too broad (which is not surprising, since there has been little thought from the review authors throughout the text). Which approach(es) listed in the review the authors consider the most promising? most questionable and needing verification? and why? An important issue which has been completely missed is that although it is useful to separate the battery into parts to carry out a focused research, the whole battery (which is an ultimate goal in this field) is not just a sum of the components. Are there any signs that this or that type of electrolyte, being superior as such, performs worse in combination with particular type of anode/cathode? Or, vice versa, maybe some combination of otherwise moderate cathode, anode, and electrolyte can suddenly provide an excellent battery when assembled? Or the components can indeed be improved separately? For me, this question (which can be answered upon accurate comparative analysis of the data cited by the authors) is even more important than actual capacities, numbers of cycles, charge rate, etc achieved in the original papers.

To sum up, unfortunately at this point I cannot consider this a nice review, but rather a compiled collection of original papers divided into three groups. Therefore, my recommendations to the authors are as follows:

Response: We thank the reviewer for improving the quality of our manuscript. 

  1. Decrease the number of citations or extend your own discussion (especially in the introduction - there are 28 references per 37 lines of text - which means that there are hardly 10-15 words explaining each reference!)

Response: Thanks for your suggestion. We have decreased the number of references in the revision.

  1. Consider adding more appropriate discussion of each of the examples cited - or at least the most important ones. Include more generalization and provide your own opinion regarding the most promising/facile/cheap approaches to be considered in future research.

Response: Thanks for your suggestion. We have added some discussion about electrolyte, anode material, cathode material for SIBs at low temperature in the revision. 

  1. Please be more accurate in the citation of the original results. Provide all the information needed to properly judge about the importance of the study - the most important piece is the information on (dis)charged rate missing in many of the citations.

Response: Yes. We have changed all current densities into rates in the revision.

  1. I would suggest adding the information about the rate in Figs. 5 and 9 (probably coded by the symbol size).

Response: Yes. Rates have been added in Figure 5 and Figure 9 in the revision.

  1. Remove Fig. 10 - it adds nothing important to the listing in the text. It could be relevant in an oral presentation, but it adds nothing to a paper.

Response: Thanks for your suggestion. Figure 10 has been removed and picture orders have been rearranged in the revision.

  1. Please accurately check every letter in the paper. Besides grammar errors, there are small yet inaccurate things: for example, incorrect units in line 49, 'Figured' in line 67, subscripts in lines 120 and 123, units in line 195, 'anode' instead of 'cathode' in line 287, 'former' in line 401, the name in line 440, 'imidazolyl' instead of 'imidazolium' in line 624, obsolete hyphens in the chemical names in lines 640 and 654, etc. The list is not complete, there are just examples of numerous inaccuracies. 

Response: Yes. Every letter in the paper is checked carefully and mistakes have been corrected in the revision.

  1. There are too many abbreviations, some of which are quite confusing (for example, NMP in the field of batteries conventionally stands for N-methylpyrrolidone rather than to a special type of electrode material). Moreover, some of the abbreviations are introduced well after their first use (for example, HC for hard carbon is introduced in line 308, being used in line 142). Some of the abbreviations are never explained (for example, ES in line 89). Therefore, I recommend the authors to add a separate section with the abbreviations used in the review.

Response: We are sorry for this confusion. We have given full explanation of the abbreviations when they appeared the first time in the revision.

  1. Regarding lines 577-579. I do not see how the optical images allow the conclusion on the viscosity. Please explain in more detail.

Response: Yes. We explained how the optical images allow the conclusion on the viscosity in the revision (Page 19 Line 35-39).

Round 2

Reviewer 2 Report

I see that the authors have addressed the weak (in my opinion) points of their manuscript. Maybe it has not turned ideal (again, in my opinion), but it has been much improved and now seems acceptable. There are still small things regarding the style and formatting (like dashes instead of minus signs) but I hope most of them can be fixed during typesetting. Anyway, small glitches are unavoidable in such long text, but the manuscript overall looks at least understandable.